# Automated navigation of condensate phase behavior with active machine learning

Yannick H. A. Leurs[1,2], Willem van den Hout[1,2], Andrea Gardin[1,2], Joost L. J. van Dongen[1], Andoni Rodriguez-Abetxuko [1], Nadia A. Erkamp[1], Jan C. M. van Hest [1]✉, Francesca Grisoni [1]✉ & Luc Brunsveld [1]✉

Biomolecular condensates are essential cellular structures formed via biomacromolecule phase separation. Synthetic condensates allow for systematic engineering and understanding of condensate formation mechanisms and to serve as cell-mimetic platforms. Phase diagrams give comprehensive insight into phase separation behavior, but their mapping is time-consuming and labor-intensive. Here, we present an automated platform for efficiently mapping multi-dimensional condensate phase diagrams. The automated platform incorporates a pipetting system for sample formulation and an autonomous confocal microscope for particle property analysis. Active machine learning is used for iterative model improvement by learning from previous results and steering subsequent experiments towards efficient exploration of the binodal. The versatility of the pipeline is demonstrated by showcasing its ability to rapidly explore the phase behavior of various polypeptides, producing detailed and reproducible multidimensional phase diagrams. The self-driven platform also quantifies key condensate properties such as particle size, count, and volume fraction, adding functional insights to phase diagrams.

Organization and compartmentalization are fundamental aspects of nature[1]. The spatial arrangement of biomolecules is essential for maintaining cellular function and facilitating metabolic processes, such as molecular transport, energy production, and structural support[2,3]. In this respect, biomolecular condensates have gained significant interest in recent years, as these membrane-less organelles play essential roles in compartmentalization and may contribute to the emergence of cellular complexity[4,5]. Condensates are phase-separated, micron-sized subcellular droplets that are formed through multivalent interactions between (macro)molecules, such as proteins and nucleic acids[6]. Their dynamic formation mechanism and complex biochemistry have become topic of intensive investigation, in particular in the field of molecular and cell biology[7,8].

An alternative approach to provide valuable insight into these structures is to engineer synthetic condensates in vitro—outside of the cellular environment[9–14]. This allows a more systematic tuning and study of the physicochemical properties of condensates[15] and also enables the development of self-assembled and/or cell-mimetic platforms that can be used for the exploration of novel therapeutic strategies[16–22]. Although synthetic condensates circumvent the need to take the cell's complexity into account, still significant challenges remain in terms of predicting condensate formation and properties based on the molecular structures and elucidating the effects of environmental factors, such as pH and ionic strength, on condensate formation and properties (as well as the underlying molecular mechanisms)[23,24].

However, it quickly becomes unfeasible to manually navigate the vast combinatorial space, given it spans diverse molecular structures and environmental factors. This process involves preparing hundreds to thousands of samples, each with precisely controlled conditions (e.g., concentration, pH, and ionic strength), followed by detailed and consistent analysis of the phase separation parameters[25–30]. Often,

[1]Institute for Complex Molecular Systems (ICMS), Department of Biomedical Engineering, Eindhoven University of Technology, Eindhoven, The Netherlands. [2]These authors contributed equally: Yannick H. A. Leurs, Willem van den Hout, Andrea Gardin. ✉e-mail: j.c.m.v.hest@tue.nl; f.grisoni@tue.nl; l.brunsveld@tue.nl

researchers are interested in measuring the binodal, the boundary in a phase diagram separating the single-phase region from the two-phase region. Beyond this boundary, the homogeneous phase separates into two distinct phases with equal chemical potentials. Identifying the binodal by collecting data across a broad range of conditions without specific guidance (*e.g.*, based on intuition) is not only time-consuming and labor-intensive but also prone to human error, highlighting the need for automated, machine learning-driven, high-throughput methods[31,32].

To address these challenges, recent innovations in high-throughput biochemical assays, microfluidics, and automated microscopy and analysis have enabled new methods to study biomolecular condensates under varied conditions[33–36]. Notwithstanding these advances, fully leveraging the vast datasets these techniques produce opens opportunities to explore condensate behavior more efficiently. Integrating machine learning, particularly active learning[37,38], into this field presents a valuable opportunity to enhance data-driven parameter exploration, refine predictive models, and reduce the need for extensive experimental input. Active machine learning iteratively selects the most informative data points to analyze and to steer the next iteration of experiments[39–42], which makes it particularly useful for automation by reducing the amount of data and experimentation needed to achieve accurate results[39,43].

In this work, we introduce an automated, high-throughput platform designed to map multi-dimensional phase diagrams of biomolecular condensates. Our platform integrates active machine learning for phase mapping optimization, an automated pipetting system for sample formulation, and an autonomous confocal microscope for high-content imaging and detailed sample characterization. Using this platform, we extensively examine the phase behavior of two well-studied polypeptides across a range of formulations. Beyond reproducibly identifying the binodal, the platform also measures particle size, particle count, and volume fraction—offering deep insight into condensate characteristics. To demonstrate the robustness of the approach, we construct higher-dimensional phase diagrams, allowing to uncover how multiple factors influence condensate formation. The automated platform not only accelerates and standardizes phase separation behavior mapping but also enhances our understanding of environmental parameter effects on condensate properties. We expect this approach to increase the application potential of synthetic condensates as a platform for the study of their natural analogues and to engineer self-assembled cell-mimetic platforms.

## Results

### Closed loop navigation of coacervate formation

Condensate formulation typically involves mixing complementary components, for example, of anionic and cationic nature[44], at specific speeds and durations, in a pH-controlled aqueous solution to form condensate, or more specifically, complex coacervate microdroplets (Fig. 1A). This process can be laborious, error-prone, and time-consuming, limiting current capabilities to determine detailed phase diagrams and, correspondingly, the optimal conditions for condensate

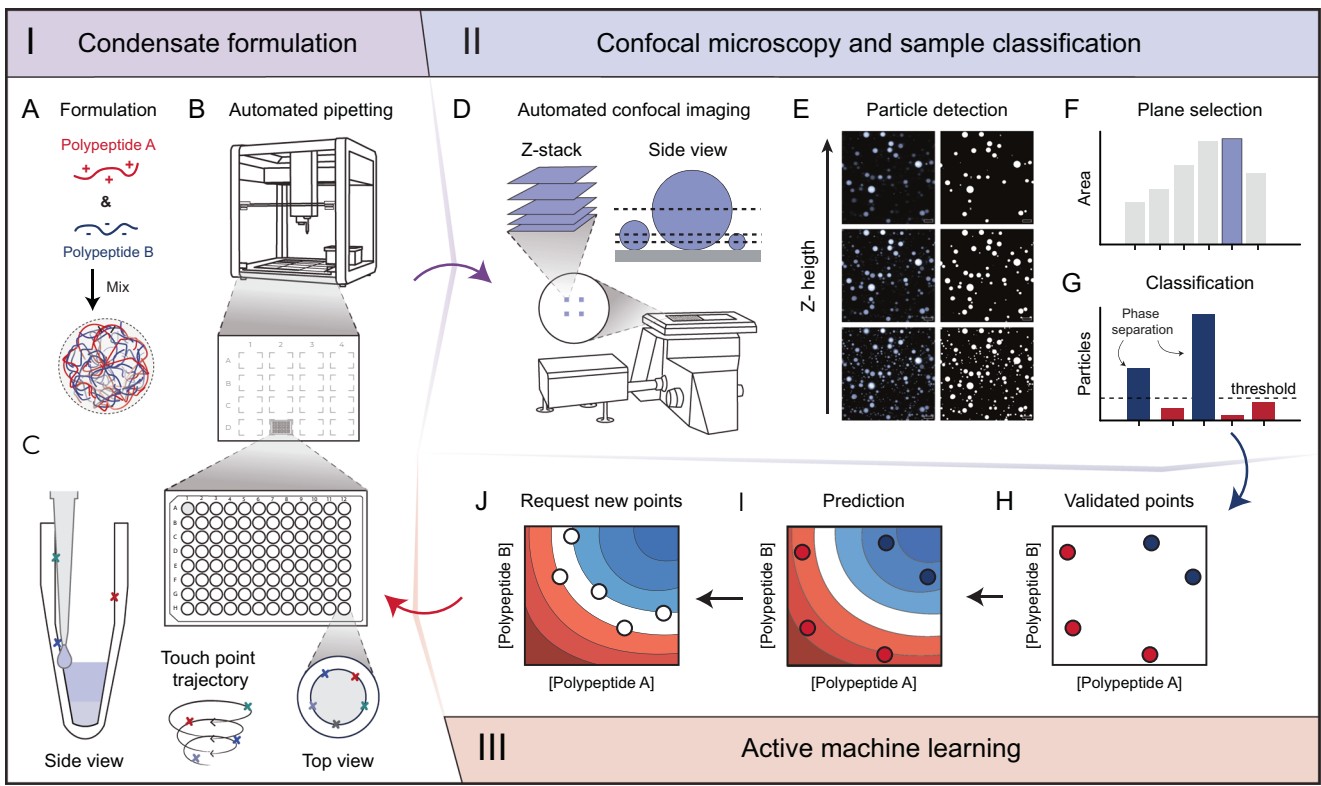

**Fig. 1 | Closed loop navigation of condensate phase diagrams.** The workflow is constituted by three parts: (I) condensate formulation, where samples are automatically prepared, (II) confocal microscopy and sample classification, for characterization, and (III) active machine learning, that learns from the collected data and suggests the next experiments. **A** Condensate microparticles are formed by mixing cationic and anionic polypeptides, resulting in phase-separated micron-sized droplets. **B** Schematic representation of the robotic pipetting platform with 16 flexible deck slots. **C** Formulations are prepared in a conical PCR plate, using contactless dispensing with volume tracking. A custom touch-tip functionality follows a touch-point trajectory to ensure accurate dispensing. **D** Confocal imaging is performed using dynamic Z-stack acquisition. **E** Example segmentation of representative confocal microscopy data using automated binary Yen-thresholding for particle detection in each Z-plane. **F** The optimal Z-plane is selected based on the largest detected area, corresponding to the slice that is best in focus. **G** Samples with 12 or more particles are labeled as phase-separated (condensates), while those below the threshold are labeled as non-condensates. **H** Experimentally validated data points are incorporated into the machine learning algorithm for training. **I** The model predicts a phase diagram based on the acquired experimental data. **J** The model then guides the selection of new formulations, restarting the automation cycle at (**A**).

formation. To date, there is no standardized protocol for producing condensates, and scientists often adhere strictly to formulation techniques that work for their specific applications[45]. Here, we present a generalizable, closed-loop workflow that combines automation and machine learning to (a) standardize and speed up condensate preparation and reduce handling errors, (b) provide an automated characterization approach, and (c) navigate complex coacervate phase diagrams more efficiently, thanks to machine learning predictions. The workflow is based on the following mutually interacting components:

I.  *Robotic sample production*. Efficient, accurate, and contamination-free sample preparation is critical for exploring vast experimental spaces with diverse conditions. Our platform addresses these needs with a cost-effective and versatile robotic pipetting platform (Fig. 1B) that combines adaptable deck space, scalable reservoir options, and an open-source programming interface. These features enable high-throughput automation of condensate formulations in any pre-defined, multi-dimensional experimental space. Custom features (Fig. 1C) allow increasing production rates through optimized liquid handling and prevent cross-contamination by using adaptable dispensing heights for contactless dispensing and different contact points for each liquid via a custom touch-tip functionality. Together, this also reduces plastic consumption, by allowing tip re-use for each distinct liquid.

II. *Automated particle characterization*. High-throughput condensate analysis requires high-throughput imaging with sufficient spatial resolution and consistent focus, which can be challenging due to heterogeneous and varying sizes of condensate sizes. In our platform, samples are transferred to a 96-well microscopy plate and imaged using an automated confocal microscope (Fig. 1D). This setup enables high-speed imaging and precise focus tracking through hardware autofocus. After formulation, condensates naturally settle over time on the glass surface, allowing for 3D reconstruction through dynamically acquired $Z$-stacks at four positions within each sample (Fig. 1D). This approach generates technical replicates and accounts for potential inconsistencies. The automated image analysis pipeline involves (a) applying binary thresholding to detect particles (Fig. 1E), (b) identifying the optimal $Z$-plane where each particle is in the best focal plane (Fig. 1F), and (c) classifying the sample as phase-separated when a threshold number of particles is observed (Fig. 1G), or as non-phase-separated otherwise (Supplementary Fig. 1). Additionally, condensate properties, such as morphology and volume fraction, are extracted for follow-up analysis and characterization.

III. *Active machine learning*. In our platform, the collected experimental data (Fig. 1H) are used to train a Gaussian Process Classifier (GPC), a machine learning model that leverages Bayesian probability to make predictions, while accounting for uncertainty in classification decisions[46]. The model is trained to predict whether a pair of polypeptides at specific concentrations (optionally along with other experimental parameters) will phase-separate. The trained model is then used to predict the phase-separation behavior of the pre-defined experimental space (Fig. 1I). Based on the predictions, new experimental points are requested for the next experimental iteration (Fig. 1J). This is achieved via the exploration of areas in the phase diagram with high prediction uncertainty (in the form of information entropy, see Methods, Eq. 5), and via diversity-based sampling (via so-called farthest point sampling[47]). The selected points are then produced and characterized (via steps **I** and **II**) and contribute to the next phase of model training.

Thanks to this closed-loop make-analyze-predict cycle, the sample production (step **I**) and characterization (step **II**) produce data for the machine-learning-driven choice of the next experiments (step **III**)—this

procedure is repeated until convergence. According to self-driving lab autonomy criteria, our pipeline qualifies as a Level 4 platform, since it integrates multiple hardware operations (e.g., liquid handling and imaging) with iterative, software-driven decision-making[48]. In this framework, the machine learning algorithm autonomously selects future experiments, and the system automatically evolves based on the newly acquired experimental data, while humans are only tasked with defining the initial search space[49]. This setup goes beyond traditional, trial-and-error based approaches, and it can generalize to virtually any system: once the initial search space is defined, the condensate phase behavior can be automatically explored in a self-driving manner.

## Proof-of-concept: automated construction of phase diagrams

To showcase the potential of our self-driving platform, we applied it to navigate the phase behavior of poly-L-(lysine) and poly-L-(aspartic acid) (Fig. 2), two well-investigated polypeptides in phase separation research[30,50–54]. Even in this case, despite their widespread use, the detailed phase diagrams that capture their binodal remain under-explored, possibly owing to the need of labor-intensive experiments[29,30,51]. In this context, this condensate system was a useful case-study to investigate how well our automated workflow was suited to effectively determine its phase behavior.

Our experiments followed the make-analyze-predict workflow, as follows:

1.  *Initialization step* (Fig. 2A). We constructed an experimental design space, ranging from 0.1 to 8.1 mM monomer concentration for each polypeptide. As a starting point we used poly-L-(lysine)$_{100}$ and poly-L-(aspartic acid)$_{200}$. Eight points for the experimental formulation and characterization were selected by the farthest point sampling algorithm[47], which starts from a randomly selected point, and then chooses maximally dispersed samples across the design space.

2.  *Automated sample production and characterization*. The chosen samples were then formulated and characterized experimentally for their phase separation (Fig. 2B, Supplementary Fig. 2). Based on their phase separation behavior, they were labeled as either 'condensate', or 'non-condensate' for training the machine learning model.

3.  *Model training and experiments selection*. The experimentally determined labels were used to train the model and predict the coacervate behavior across the design space (Fig. 2C). In particular, the GPC algorithm generates a new phase diagram prediction across the design space. The probabilistic nature of GPC prediction allows to compute an uncertainty measure per class, which we leverage in the form of entropy of the class probabilities (the higher the entropy, the higher the uncertainty across the classes, see "Methods", Eq.( 4)). Once the points within the highest uncertainty regions are identified, farthest point sampling again selects the next batch points for production and characterization (Fig. 2D).

After the initialization (step 0), steps 1–2 were iteratively repeated, by adding the new experimental labels to the training dataset and subsequently updating both the phase (Fig. 2E) and uncertainty (Fig. 2F) landscapes for the next cycle. This active learning process continued until a total of 72 samples were measured across nine cycles (Fig. 2A, G).

After approximately 40 samples (five iterations), only minor changes were observed in the predicted phases, suggesting that the model started to stabilize. Collecting a total of 72 samples further reduced the uncertainty of the predicted phase boundaries (Supplementary Figs. 3–5). To minimize nonspecific surface interactions, all experiments were performed in BSA-coated plates, which we verified to have no detectable influence on phase behavior (Supplementary Fig. 6). The automated exploration of the phase diagram was carried

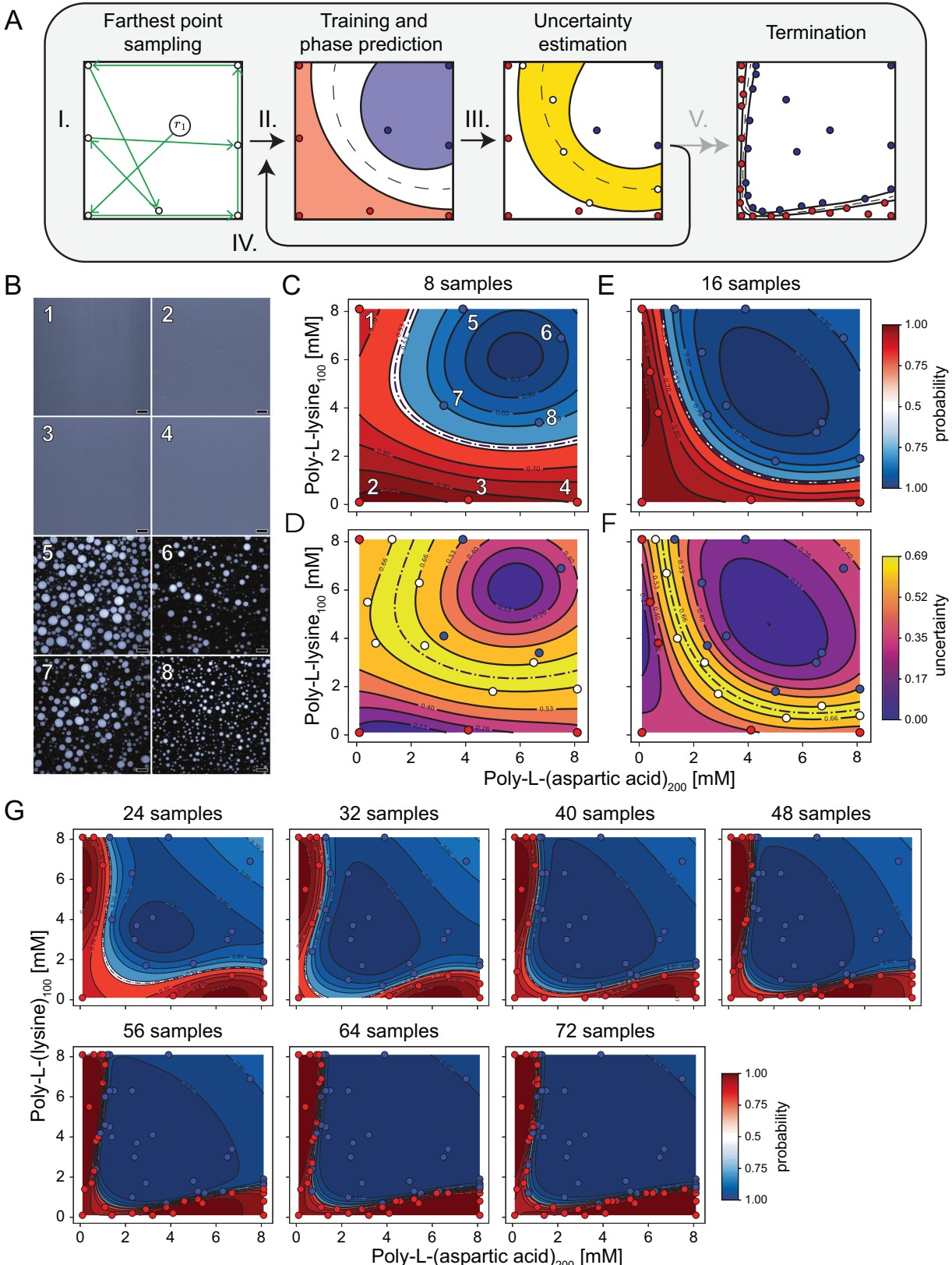

out in approximately four hours, whereas conducting these experiments manually would have required more than one week. Additionally, the active learning approach generated a detailed phase diagram, a result that would have otherwise required the intuition of an experienced scientist to achieve manually, and potentially many more datapoints (Supplementary Fig. 7). As a further validation, we tested

the pipeline on a synthetic phase diagram containing multiple isolated negative phases (Supplementary Fig. 8). The model successfully identified these hidden regions, demonstrating its flexibility and robustness in navigating complex phase landscapes. Together, these results highlight the platform's effectiveness in reducing time and guiding experimental efforts toward the most relevant areas.

**Fig. 2 | Machine-learning guided condensate 2D phase diagram mapping.**
**A** Schematic of the active machine learning pipeline used for phase diagram mapping. (I) The initial sample points are selected using farthest point sampling to ensure broad coverage of the design space. (II) A Gaussian Process Classifier is trained on the data, generating a preliminary phase diagram. (III) An uncertainty landscape is computed, highlighting regions with the highest uncertainty. From these regions, new points are sampled using farthest point sampling. (IV) The selected samples are experimentally validated and added to the dataset, refining the phase diagram prediction. (V) Steps I–IV are repeated until convergence is achieved. **B** Representative confocal micrographs for the first eight experimentally validated samples (scale bar = 20 μm). **C** The predicted phase diagram for poly-L-(aspartic acid)$_{200}$ and poly-L-(lysine)$_{100}$ based on the validated samples in (**B**). Phase separation is represented by blue points and no separation by red points, with the surface depicting the model's predictions. **D** The entropy landscape is constructed based on the prediction in (**C**), and new samples (white points) are selected using farthest point sampling in the high entropy region of the landscape. The requested points are experimentally classified, and a new phase diagram is predicted from the combined data (**E**), along with its associated entropy landscape (**F**). **G** Subsequent iterations continue until 72 data points are acquired. Phase boundaries are indicated by dotted lines; in some cases, these may be partially obscured by overlapping contour lines. Total polypeptide consumption: 4.0 mg poly-L-(aspartic acid)$_{200}$ and 4.2 mg poly-L-(lysine)$_{100}$.

Notably, the resulting shape of the phase diagram is consistent with the physical principles underlying condensate formation. In associative LLPS, droplet formation is driven by multivalent electrostatic interactions and the release of bound solvent molecules, which together must outweigh the entropic cost of reduced chain flexibility[55,56]. These conditions are optimally met near stoichiometric charge ratios, where attractions between the polypeptides are maximized. When one component is in excess or depleted, the resulting charge imbalance and electrostatic screening hinder the formation of an extended interaction network, thereby suppressing phase separation as reflected in the mapped binodal[57]. Additionally, turbidity and DLS measurements support the observed phase boundary (Supplementary Fig. 9), although DLS also detected the formation of nanometer-scale assemblies below the resolution limit of confocal microscopy.

## Convergence of condensate phase mapping

A desirable feature when automatically mapping phase diagrams is the unified convergence and reproducibility of the final results regardless of the starting points selection. In fact, while the designed space available for selecting experimental conditions is vast (in this study, a grid of 6561 points), the models are trained in a low-data regime (up to 72 datapoints), which opens questions about how the underlying patterns and trends are captured[58]. Moreover, given the iterative nature of the approach, initial decisions (e.g., starting points for training) and automation-related challenges (e.g., equipment inconsistencies) might affect decisions in later cycles. To shed light on this key question, we performed three independent replicates using the poly-L-(lysine)$_{100}$ and poly-L-(aspartic acid)$_{200}$ system, so that each replicate was carried out identically (as explained above), but starting from a unique and non-overlapping initial set (step 0) for model training (Fig. 3A).

Since the starting sets highly differed across replicates, they resulted in different phase and uncertainty landscapes in early cycles (Fig. 3B, Supplementary Figs. 10–13). While each run followed its 'prediction route' across cycles, after approximately 40 samples (cycle number 5), the phase diagrams appeared to converge across the replicates. After collecting 72 samples (cycle number 9), the replicate phase diagrams displayed remarkable similarity and low uncertainty levels.

To further assess the reproducibility of our experiments, we constructed a "ground truth" phase diagram (Supplementary Fig. 14) using all data collected across replicates (Supplementary Figs. 15–20). We quantified the prediction agreement between each replicate's predictions (at each cycle) and the ground truth via balanced accuracy (the higher, the more similar the predictions, see "Methods" Eq.(7))[59]. Across replicates, the balanced accuracy steadily increased over successive cycles (Fig. 3C), which is especially visible from the fifth cycle onwards, where balanced accuracy reached values consistently above 95% across all replicates. This indicates that, no matter the starting point, all replicates converge to a similar phase diagram in a data-efficient way (*i.e.*, by using substantially less data than the "ground truth" diagram). These results agree with existing active learning literature[39,58,60], showing the potential of this approach to progressively mitigate the effect of the starting data.

The Jensen-Shannon divergence[61] (see "Methods", Eq.(8), (9)) was computed to directly compare phase diagrams (the lower the divergence, the more similar). A "within-replicate" divergence was calculated, by comparing the predicted probabilities of each replicate across consecutive cycles (Fig. 3D). The results showed an exponential decrease in divergence values, with substantial changes in the predicted phase diagrams within the first 32 samples (cycle 4) and minimal changes after 56 samples (cycle 7), suggesting that each phase diagram reached a 'stable' state, where additional experiments did not significantly alter predictions. Moreover, we calculated a "between-replicate" divergence (Fig. 3E), by comparing the predictions of each cycle across different replicates. The divergence values decreased sharply during the first three cycles, after which they stabilized. These results indicate that only three cycles were necessary to mitigate the stochastic differences by the different starting points, after which the replicates progressively aligned along a common trajectory.

## Mapping condensate properties via phase diagram exploration

Traditional studies on phase separation behavior have primarily focused on determining whether condensates form under specific conditions[27,28,30,34,62]. Our automated data production and characterization pipeline collects additional information beyond phase separation. In particular, depth-resolved imaging from confocal microscopy allows to derive several properties of condensates, including particle count, morphology, and volume fraction, within the phase diagram. Here we compounded the data from the previously described replicates, along with data obtained from optimization experiments, totaling 480 experimentally determined samples (Fig. 4A, Supplementary Fig. 14). The collected samples show a wide range in particle morphologies, ranging from densely packed condensate clusters to tiny, barely visible particles (Fig. 4B). This diversity underscores the variability in condensate formation even within a single "simple" phase, underscoring the necessity of collecting a broader set of properties to gain a deeper understanding of phase behavior.

Here, we focused on the following condensate properties: (a) number of detected condensates, (b) average particle area, and (c) total volume fraction, extrapolated by combining particle counts and area. These properties were mapped onto the compounded phase diagram, and all of them showed evident trends across the experimentally determined space. Low particle counts were for example observable near phase boundaries, while the count increased when both protein concentrations increased (Fig. 4C). Particle size showed a similar trend, with larger condensates forming at higher concentrations (Fig. 4D). Some regions showed fewer but larger particles, suggesting potential fusion (coalescence) of condensates due to surface saturation. Volume fractions were lower near the binodal and higher toward the inner part of the phase separated region (Fig. 4E), in line with complementary Nanoparticle Tracking Analysis (NTA) measurements (Supplementary Fig. 21).

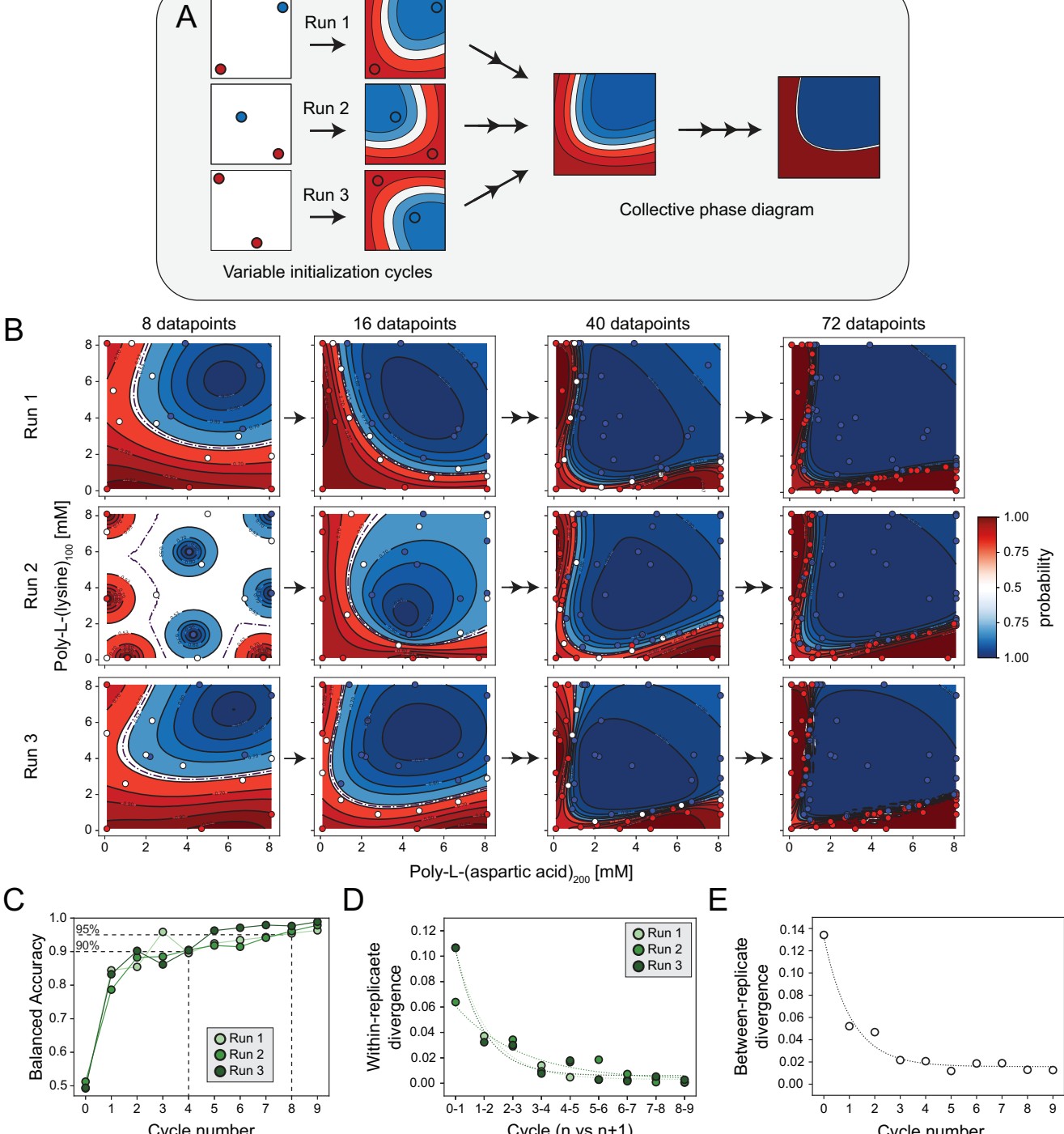

**Fig. 3 | Consistent convergence of phase mapping across replicates.**
**A** Schematic illustration of the experimental workflow used to produce replicated phase diagrams. The initial set of experimental samples is selected by farthest point sampling, resulting in different starting points for each replicate. Each subsequent cycle then follows a unique path to reach the same phase diagram. Reproducibility across replicates is expected only if the machine learning, formulation, and analysis steps are consistent. **B** Phase diagrams, showing phase separation in blue and no separation in red, with probability prediction fits (background), validated points (colored dots), and new sample selections (white dots) for three experimental replicates of poly-L-(lysine)$_{100}$ and poly-L-(aspartic acid)$_{200}$ condensates. A total of 72 data points is experimentally validated across 9 cycles. Representative cycles are shown, remaining cycles and entropy maps are reported in Supplementary Figs. 4, 5 and Supplementary Figs. 10–13. **C** Balanced accuracy plot showing the accuracy on

the prediction for each successive cycle with respect to a "ground truth" phase diagram. Cycle 0 represents the balanced accuracy computed with respect of a randomly generated phase diagram as a baseline comparison. **D** Average Jensen-Shannon Divergence plot illustrating within-experiment divergence by comparing consecutive cycles for each replicate. This reflects the progressive convergence toward the final phase diagram for each replicate. Cycle 0 is a random phase diagram included as a reference for low similarity. **E** Average Jensen-Shannon Divergence plot comparing divergence across replicates at each cycle, highlighting inter-experiment variation. Cycle 0 compares three random phase diagrams and is included as a reference for low similarity. Phase boundaries are indicated by dotted lines; in some cases, these may be partially obscured by overlapping contour lines. Polypeptide consumption per phase diagram: 4.0 mg poly-L-(aspartic acid)$_{200}$ and 4.2–4.8 mg poly-L-(lysine)$_{100}$.

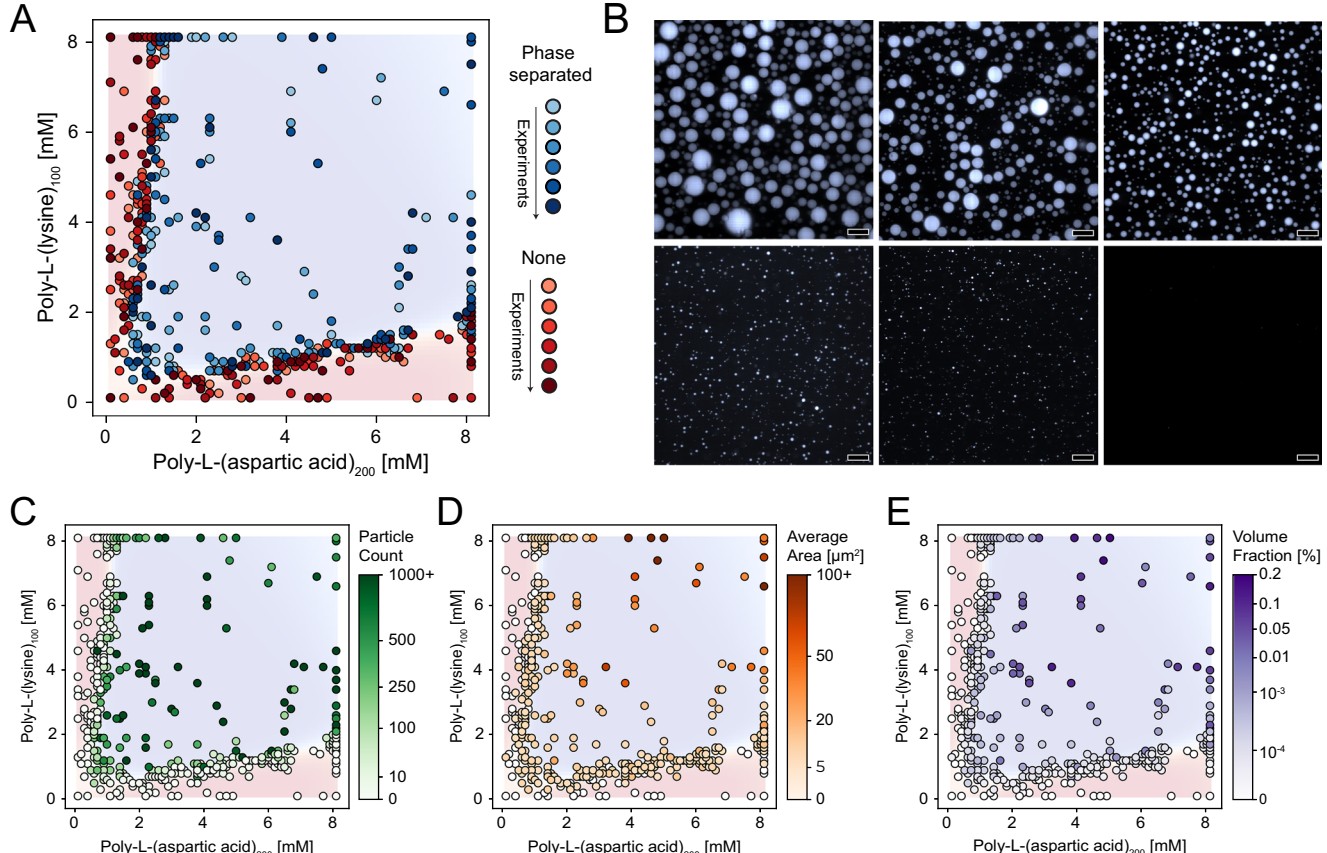

**Fig. 4 | Condensate properties beyond phase boundaries. A** Combined 2D phase diagram of poly-L-(lysine)₁₀₀ and poly-L-(aspartic acid)₂₀₀, based on 480 validated data points. The data is compiled from six independent experiments, represented by varying shades of blue (indicating phase separation) and red (indicating no phase separation). **B** Representative confocal micrographs that illustrate the wide range of observed condensate phenotypes (scale bar = 20 μm). Each image corresponds to a different sample condition; micrographs are shown to demonstrate morphological diversity rather than biological replicates. (**C**–**E**) Quantification of condensate properties overlaid on the phase diagram in (**A**): (**C**) number of detected condensates, (**D**) average particle area, and (**E**) apparent volume fraction, estimated from particle count and size. White points indicate conditions where no particles were detected. Total polypeptide consumption for 480 samples: 26.4 mg poly-L-(aspartic acid)₂₀₀ and 29.8 mg poly-L-(lysine)₁₀₀.

Notably, our property measurements represent standardized snapshot observations taken after a 15-min incubation period of a dynamic, continuously evolving system. While the incubation time can be adjusted to suit the user's objective or extended to enable kinetic measurements (Supplementary Figs. 22–24), we selected 15 min as a practical and reproducible readout, based on DLS and turbidity data showing that key features for identifying phase separation begin to stabilize around this time (Supplementary Fig. 25). This approach enables systematic mapping of phenotypic variations across the phase diagram, offering insights into condensate behavior beyond the binary presence or absence of phase separation. To further extend the platform's scope, we also performed preliminary measurements of dense-phase concentrations (Supplementary Fig. 26), providing a basis for future composition-dependent analyses that could approximate partitioning coefficients and the associated thermodynamic driving forces[63]. By enabling broad, quantitative screening, these metrics lay the groundwork for deeper, targeted analyses using advanced biophysical tools (i.e., FRAP, microrheology, or optical trapping) to probe material properties such as viscosity, dynamics, or mechanical behavior. Collectively, these quantitative descriptors may provide a basis for integration with theoretical models of phase behavior, while also enabling more informed and targeted formulation strategies, particularly in regions of the experimental space where specific material properties rather than phase separation alone are critical for function or downstream application[64–66].

**Identifying structure-separation relationships with automation**

To further extend the applicability of our workflow, we applied it to elucidating how polypeptide chain length affects phase separation. We constructed phase diagrams for nine combinations of poly-L-(lysine) and poly-L-(aspartic acid) polypeptides, each differing in chain length (poly-L-(lysine)ₙ: $n$ = 20, 100, 250; poly-L-(aspartic acid)ₙ: $n$ = 30, 100, 200) but with constant overall monomer concentrations. All combinations exhibited phase separation within the tested experimental space (Fig. 5, Supplementary Figs. 27–42). However, although these polypeptides share the same structural monomeric unit, their phase behavior, as well as their properties (Supplementary Figs. 43–45) varied considerably. The machine-learning-guided exploration of these phase diagrams was carried out in approximately one week, whereas conducting these experiments manually and based on intuition would have been seriously challenging and labor-intensive.

Notably, even with the more complex and curved diagrams of some of the combinations, we successfully identified well-defined phase boundaries within 72 samples (9 cycles) for all tested conditions. Generally, increasing the length of one polypeptide while keeping the length of the other polypeptide constant enabled phase separation at lower concentrations for the elongated polypeptide, but it required higher concentrations of the fixed-length polypeptide, as visible, for instance, in the case of poly-L-(lysine)₂₀ (Fig. 5A–C). Similarly, when the poly-L-(lysine) length increased from 20 to 100 or 250 repeats (Fig. 5D–I), while maintaining a constant poly-L-(aspartic acid) length,

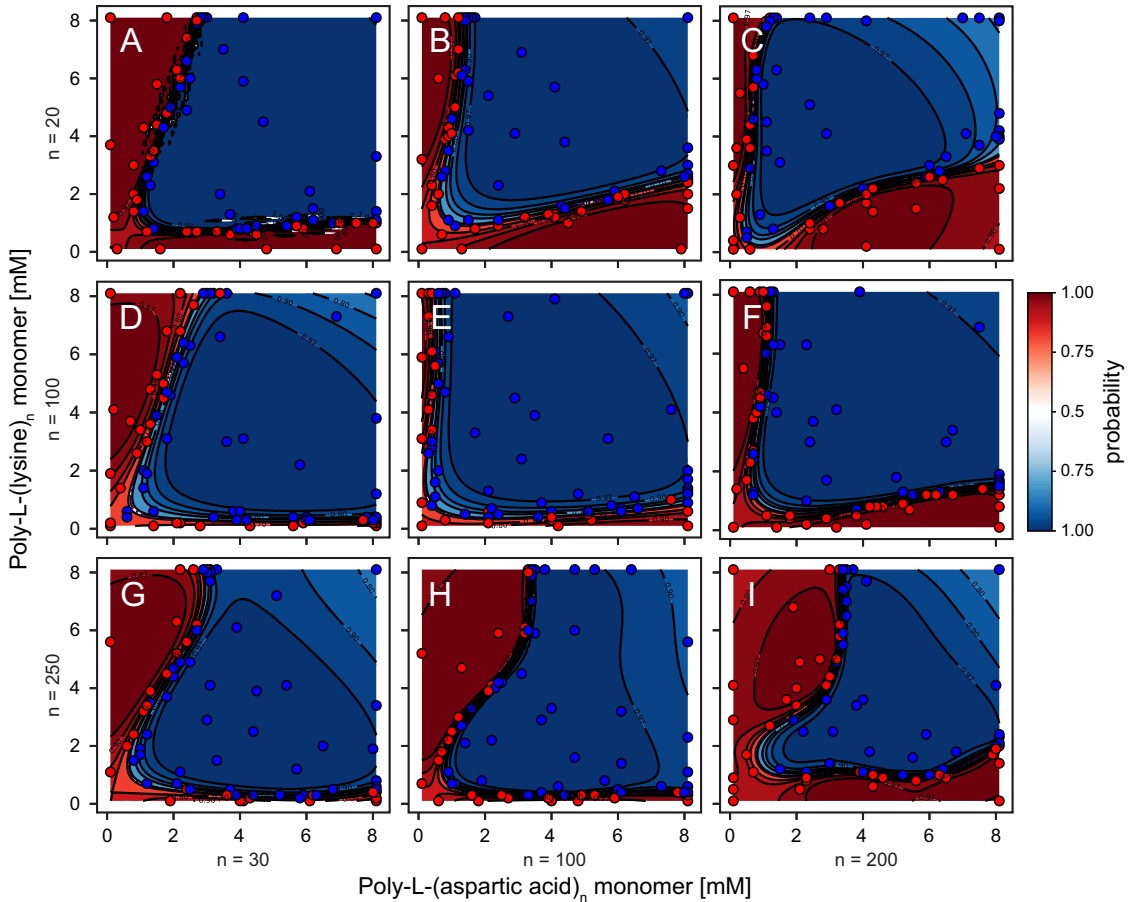

**Fig. 5 | Impact of polypeptide length on phase separation behavior.** This figure displays nine phase diagrams illustrating the automated mapping of phase separation for combinations of poly-L-(lysine) and poly-L-(aspartic acid) with varying chain lengths. Panels represent phase diagrams for poly-L-(lysine) with a chain length of 20, combined with poly-L-(aspartic acid) of lengths 30 (**A**), 100 (**B**), and 200 (**C**). Panels show poly-L-(lysine) with a chain length of 100, paired with poly-L-(aspartic acid) lengths of 30 (**D**), 100 (**E**), and 200 (**F**). Panels (**G-I**) depict poly-L-(lysine) with a chain length of 250, combined with poly-L-(aspartic acid) lengths of 30 (**G**), 100 (**H**), and 200 (**I**). Datapoints are marked as dots, with blue indicating phase separation and red indicating no phase separation. Each phase map includes a background color gradient derived from predictions based on 72 datapoints per combination, acquired over nine cycles of eight datapoints. Remaining cycles and entropy maps are reported in Supplementary Figs. 27–42. Phase boundaries are indicated by dotted lines; in some cases, these may be partially obscured by overlapping contour lines. Total polypeptide consumption per phase diagram: 4.0–4.9 mg poly-L-(aspartic acid) and 3.9–5.4 mg poly-L-(lysine).

phase separation occurred at lower lysine concentrations, but required higher concentrations of poly-L-(aspartic acid).

These results highlight the delicate balance required in designing polypeptide systems for phase separation. Simply increasing the concentration or length of one polypeptide does not necessarily lead to enhanced phase separation; instead, the process is highly sensitive to the interplay between both polypeptides. Our findings indicate that an optimal balance exists at equal chain lengths of 100 repeats (Fig. 5E), where phase separation occurs extensively across most of the investigated chemical space. In some cases, particularly with poly-L-(lysine)₂₅₀, phase boundaries showed slight bends, suggesting complex, non-linear dynamics. These complexities highlight the challenges in controlling and predicting condensate formation, as even minor adjustments at the molecular level can lead to pronounced changes in phase behavior.

### Navigating phase behavior in complex environments
Building upon these results, we increased experimental complexity by introducing salt (NaCl) as an additional dimension to our system. Salts modulate electrostatic interactions between charged polypeptides and thereby significantly influence condensate phase behavior and properties[24,67]. This expansion increased the potential experimental space from 6,561 points (two dimensions) to 531,441 points (three

dimensions). To evaluate the platform's performance, we performed two independent replicates using the poly-L-(lysine)₁₀₀ and poly-L-(aspartic acid)₂₀₀ system for 20 active learning cycles with 32 samples each (640 measured points per replicate; Supplementary Figs. 46–47). These newly acquired points were compounded with previous data to construct a comprehensive 3D "ground truth" phase diagram (Fig. 6A, B). As anticipated, salt greatly influenced condensate formation, promoting phase separation at moderate concentrations (150–700 mM), while disrupting it at higher concentrations (1200–1300 mM)[45]. Interestingly, some phase-separated regions at higher salt concentrations were identified (Fig. 6B, 270° rotation), which result from salt-induced aggregate phases (Supplementary Fig. 48). As our current analysis detects any contiguous fluorescent signal above the pixel threshold, these aggregates were classified as 'phase separated', regardless of internal structure or material state.

To assess the pipeline's reproducibility and performance, we again calculated the balanced accuracy[59] (See "Methods", Eq.( 7), Fig. 6C) and within- and between-replicate Jensen-Shannon divergence[61] (see "Methods", Eqs.( 8), (9), Fig. 6D, E). As anticipated, all metrics showed consistent improvements across cycles and rapid convergence toward the global phase diagram, with stabilization occurring after approximately eight cycles (256 samples). Notably, these metrics effectively captured the overall progression in identifying phase behavior but may

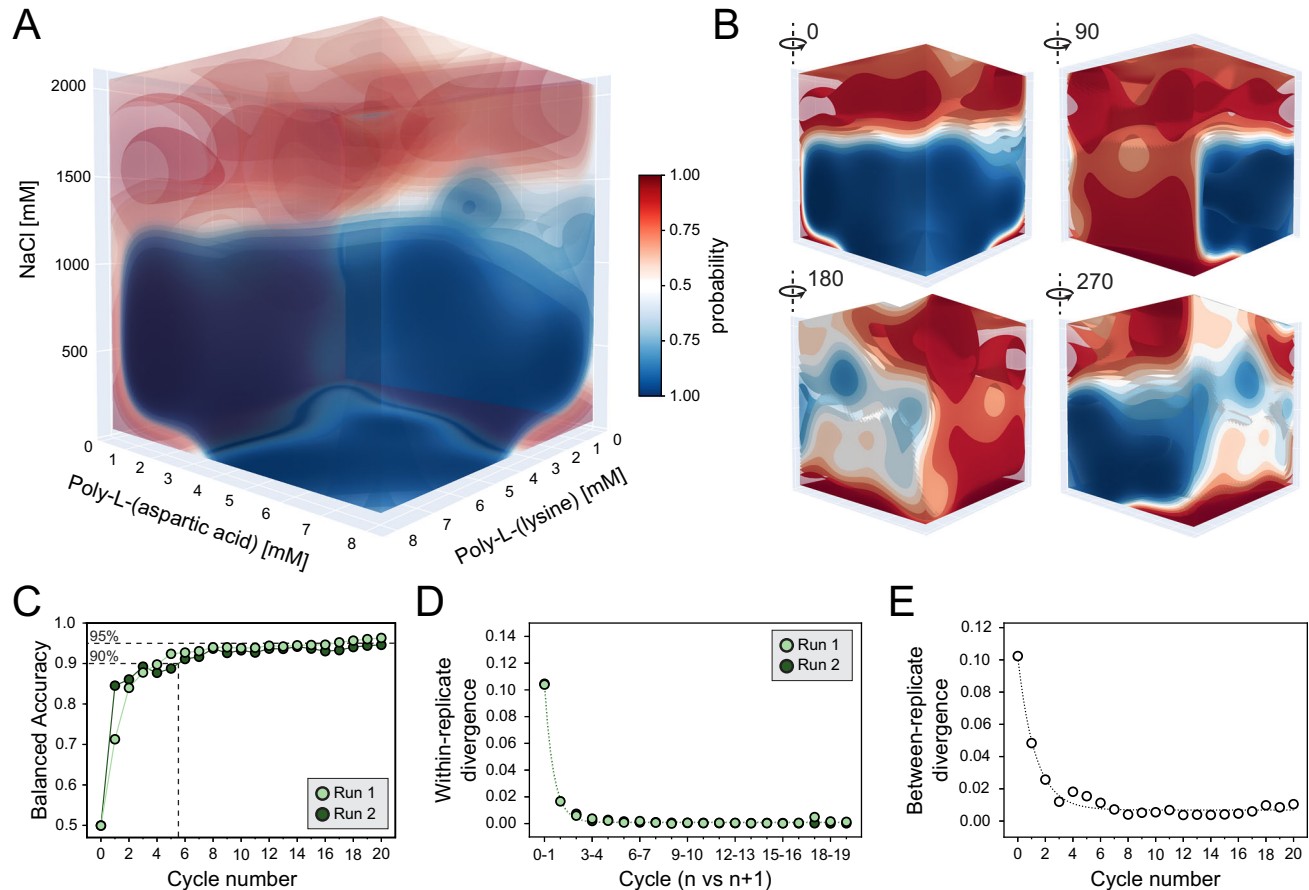

**Fig. 6 | Automated mapping of multi-dimensional phase diagrams. A** Two independent experiments (Supplementary Figs. 46–47) were conducted to explore the effect of salt (NaCl) on the phase behavior of poly-L-(lysine)$_{100}$ and poly-L-(aspartic acid)$_{200}$. The combined dataset, made from 1760 datapoints, was used to construct the "ground truth" three-dimensional phase diagram, here reported. Iso-probability surfaces indicate phase separation (blue, higher opacity) and no phase separation (red, lower opacity). **B** Four distinct orientations of the phase diagram with non-transparent surfaces are shown to emphasize phase behavior from different perspectives. **C** Balanced accuracy plot showing the accuracy on the prediction for each successive cycle with respect to the "ground truth" phase diagram

in panel A. Cycle 0 represents the balanced accuracy computed with respect of a randomly generated phase diagram as a baseline comparison. **D** Within-experiment Jensen-Shannon Divergence (JSD) plotted across cycles. This metric tracks convergence by comparing consecutive cycles, illustrating how each replicate approaches the final phase diagram. Cycle 0 reflects divergence from a randomly generated phase diagram. **E** Between-experiment Jensen-Shannon Divergence (JSD) across replicates at each cycle. Similar to panel **D**, Cycle 0 serves as a baseline, representing divergence from a randomly generated phase diagram. Total polypeptide consumption for 1280 samples: 85.1 mg poly-L-(aspartic acid) and 82.1 mg poly-L-(lysine).

be less sensitive to minor changes in the large design space during the later stages of optimization. Nonetheless, the balanced accuracy continued to improve slightly in subsequent cycles, primarily enhancing the resolution around the phase boundaries (white areas in Fig. 6A, B; Supplementary Figs. 46–47).

Increasing dimensionality introduces challenges, both for machine learning algorithms and due to the formation of distinct aggregate phases. Despite these challenges, we successfully mapped these 3D phase diagrams in just three days. These results not only demonstrate the platform's capability to rapidly explore vast and complex design spaces but also highlight the essential role of machine learning in effectively navigating and elucidating such high-dimensional complex assemblies (Supplementary Fig. 7). To accommodate complex chemical spaces, users can flexibly adjust the resolution of the search space depending on their objectives. This is supported by simulations with down-sampled grids, which showed that early-cycle performance remains robust in 2D and 3D even with substantially reduced design spaces (Supplementary Fig. 49).

## Discussion

In this work, we presented a versatile, machine learning-driven automated platform that rapidly navigates multi-dimensional phase

diagrams of condensates. By integrating (a) active machine learning to optimize sample selection and phase diagram navigation, (b) automated pipetting for precise sample formulation, and (c) advanced and automated confocal microscopy for high-content particle characterization, we examined the phase behavior of polypeptides across various formulations and concentration profiles. Our platform reliably and rapidly identified phase boundaries with high accuracy and reproducibility, demonstrating the robustness of our approach. Additionally, it quantified key condensate properties, such as morphology, particle count, and volume fraction, providing insights beyond the traditional binary classifications of phase separation. Moreover, the platform's flexibility enabled rapid exploration of complex phase spaces, allowing to reveal the influence of polypeptide chain length and salt on phase behavior.

Looking forward, numerous opportunities exist to further enhance our platform's capabilities and broaden its applications. By refining the sampling strategies (*e.g.*, by balancing exploration of uncertain regions with exploitation of high-certainty points) the efficiency of phase diagram navigation could be further improved[68]. Furthermore, integrating robotics to enhance platform autonomy[69,70] and leveraging machine learning for advanced image analysis can significantly improve condensate classification[71]. Moreover, integrating condensate properties

into active machine learning algorithms will allow us to incorporate desirable particle properties in the decision-making process, supporting the design of biomaterials for applications such as drug delivery and tissue engineering. In the future, incorporating molecular information into machine learning models (e.g., via deep learning[72,73]) will enable linking molecular structure with phase behavior, extending beyond the training sets[74]. Finally, the platform's modularity and adaptability make it generalizable to other complex micron-sized assemblies, such as tactoids[75] and microgels[76]. Looking ahead, complementary tools such as fluorescence recovery after photobleaching (FRAP), microrheology, partitioning measurements, or the use of structure-sensitive dyes (e.g., ThT, Amytracker) could be incorporated to quantify condensate diffusivity, viscosity, molecular enrichment, or the presence of β-structured aggregates, further deepening functional insights[77,78]. This versatility also opens up opportunities to explore how minor structural and compositional changes in natural proteins, resulting from processes like splicing, mutations, and post-translational modifications, influence condensate behavior, offering valuable insights into phase separation principles under diverse conditions[79].

## Methods

### Preparation and dye labeling of polypeptides

All polypeptides used in this study were purchased from Alamanda Polymers. They were dissolved in fresh Milli-Q water (MQ) at 25 mg/mL, then sterile-filtered through a 0.2 μm filter, and stored in aliquots at −20 °C. Further dilutions were prepared in MQ, with stock solutions maintained at 4 °C.

A portion of the poly-L-(lysine) polypeptides was labeled with NHS-Sulfo-Cy5 dye (Lumiprobe) for confocal imaging. The dye was dissolved in DMSO at a concentration of 10 mg/mL and stored at −20 °C. Poly-L-(lysine) was labeled in a reaction buffer consisting of 100 mM HEPES (pH 8.0) and 150 mM NaCl in MQ. The polymer-to-dye ratios were 1:3 for poly-L-(lysine) with a chain length of 100 and 1:6 for chain lengths of 20 and 250. The reaction was carried out for two hours at room temperature while shaking at 550 rpm using an Eppendorf MixMate.

Unbound dye was removed using a PD Minitrap G-25 size exclusion column (Cytiva), which was pre-equilibrated with a storage buffer of 25 mM HEPES (pH 7.4) and 100 mM NaCl. Labeling was, if possible, further confirmed by analyzing the flow-through of the dye-labeled polymer after centrifugation with a 3 kDa spin filter (Amicon). All polypeptides were freeze-dried, weighed, and dissolved in MQ. The dye concentration was determined using a nanodrop spectrophotometer (Thermo Scientific NanoDrop 1000). This measurement, combined with the dry weight of the polypeptide, allowed for the calculation of the Degree of Labeling (DoL). The final dye-labeled polypeptides were sterile-filtered (0.2 μm) and stored at −20 °C, with additional dilutions prepared in MQ and maintained at 4 °C.

### Preparation microscopy plates

For confocal imaging, black 96-well glass-bottom microscopy plates (Cellvis, 1.5, P96-1.5H-N) were used and the glass surface was passivated to prevent wetting of the condensates. To prepare the surface coating, bovine serum albumin (BSA) was dissolved in MQ at 30 mg/mL and then sterile-filtered through a 0.2 μm filter. A volume of 100 μL of this BSA solution was added to each well. The plates were placed on a MixMate shaker (Eppendorf) and incubated at 500 rpm for 60 min at room temperature. After incubation, the BSA solution was discarded, and each well was rinsed three times with 100 μL of MQ water. The plates were then dried overnight, covered with a Kimwipe, and stored at room temperature under a protective cover until use.

### Data architecture and general automation workflow

All devices were integrated within a local network and regulated through a central orchestrator workstation, which served as the control hub for the entire platform. The orchestrator contains all necessary protocols and information, and coordinates all device actions and data exchange. Communication with platform components was achieved through USB connections and a local Ethernet network, using TCP-based network communication protocols such as SSH and HTTP.

A centralized data architecture was implemented to manage knowledge transfer between instruments. This architecture included a structured folder system on the orchestrator workstation for organizing Python protocols, instrument logs, raw data storage, and dedicated information transfer files. These information transfer files, detailed below, contained specific instructions for each device—often generated through machine-learning algorithms—and were sent from the orchestrator workstation to individual components. Each device passively listened to the orchestrator workstation, which assigned tasks and actions directly. Devices executed only the actions directed by the orchestrator, forming a streamlined, centralized data workflow across the platform.

**Master file.** Central, continuously updated database for all sample details, conditions, and results. It logs sample locations and barcoded plates, directing sample creation, handling, and analysis. A versioned copy was made before each update to maintain data integrity.

**Barcode file.** Output from machine learning, which is cross-referenced with the Master File to identify samples to be processed.

Batch File: Contains a detailed description of polypeptide stocks, including date, version, and degree of labeling for dye-labeled polypeptides. It is essential for calculating component volumes in sample preparation.

Source File: Tracks materials that are stored in a 96-well plate, including their concentrations and volumes. This file was updated after each pipetting step and versioned once per automation cycle to support accurate records.

This system operated in a closed-loop workflow, where each action depended on information from previous steps, all coordinated by the central orchestrator workstation. The workflow began with the machine learning model, which assessed the chemical space and determined the next set of samples to be measured. It appended these new sample conditions to the Master File and created a matching Barcode File. Next, the pipetting platform used information from the Master, Source, and Batch Files to calculate the required volumes and assign target locations for each sample. During sample preparation, the Source File was updated after each pipetting step to keep track of remaining volumes. Once the samples were prepared, their locations were added to the Master File. The microscope then cross-referenced the Barcode File with the updated Master File to find sample locations and imaging coordinates. It automatically acquired and processed confocal micrographs and added the classification results to the Master File. Finally, the machine learning model retrieved these updated classifications, incorporated them into the chemical space, and initiated the next cycle of experiments.

### Automated sample preparation

**Instrument setup and configuration.** Samples were prepared automatically using an Opentrons Flex pipetting robot equipped with both single- and 8-channel pipettes (5–1000 μL) and 200 μL tips. The deck was configured as follows: 200 μL tip rack in slot B1; 195 mL NEST reservoir filled with MQ in slot C2; Heater-shaker module (Gen 1) with a PCR adapter plate and either a NEST 96-well PCR plate for 2D phase diagrams or an Opentrons Tough 96-well PCR plate for 3D phase diagrams in slot D1; 2 mL 96-well deep-well plate (NEST) containing stock solutions in slot D2; waste chute in slot D3; and a 96-well microscopy plate (Cellvis) in slot C3.

**Pipette offset calibration.** The Flex platform was calibrated for height and x/y offsets, following the manufacturer's guidelines.

**Source plate setup.** Stock solutions of HEPES, NaCl, and polypeptides (labeled and unlabeled) were preloaded in the source plate (D2). The robot tracked and updated each well's volume (see Data Architecture and Workflow), prompting refills to bring wells up to 1800 μL when volumes dropped below 200 μL.

**Liquid handling.** Reagents were dispensed sequentially to achieve a final volume of 150 μL per PCR well: MQ water, HEPES buffer (50 mM, pH 7.4), NaCl (150 mM for 2D or 25–2050 mM for 3D diagrams), dye-labeled poly-L-(lysine) (96–250 nM), unlabeled poly-L-(lysine), and poly-L-(aspartic acid) (0.1–8.1 mM monomer concentration). Final calculations accounted for any additional monomers introduced by the dye-labeled poly-L-(lysine) to ensure accurate concentrations. The same tip was used for multi-dispensing reagents, with new tips used for each aspiration step (except MQ).

**Mixing.** From NaCl addition onward, samples were mixed (1500 rpm) for 10 seconds. After the final component (poly-L-aspartic acid), samples were mixed (1500 rpm) for 5 minutes to promote phase separation.

**Custom dispensing technique.** To improve accuracy, transfers used a minimum of 10 μL, leaving 5 μL of residual volume in the tip. Prior to dispensing, dynamic volume tracking adjusted the pipette height based on the anticipated liquid volume in the well. Dispensing occurred just above the liquid surface to ensure that any residual droplets hanging from the tip were reliably released into the bulk solution (Fig. 1C, side view). After dispensing, a custom touch-tip function guided the pipette to contact specific points along the well wall at the same height to remove remaining droplets. Unlike the default four-point Opentrons routine, our implementation dynamically adjusted both the number and location of contact points based on the number of distinct liquids added to each well (Fig. 1C, top view). As additional liquids were dispensed, new touch points were assigned at progressively higher vertical positions, forming a gentle upward spiral (Fig. 1C, touch point trajectory). This approach enabled accurate multi-liquid dispensing with a single tip, minimized cross-contamination risk, and maintained spatial separation between touch locations regardless of the number of liquids used.

**Final transfer for imaging.** After preparation, 100 μL of each sample was transferred to the imaging plate (C3), which was then sealed with an adhesive aluminum foil seal (ThermoFisher) for confocal imaging. A new tip was used for each well, with samples mixed three times by aspiration/dispensing before transfer. Samples were incubated for 15 minutes before analysis, unless indicated otherwise.

## Automated confocal microscopy
**Confocal microscopy setup and hardware configuration.** Imaging was conducted on a custom confocal setup integrated by Confocal NL. The microscope consisted of an open-frame inverted microscope (Zaber), with a confocal NL line re-scan system (NL5 + ) mounted on the left-side camera port. Additionally, the microscope was equipped with a motorized filter wheel (Confocal NL), and a laser autofocus module (Zaber). The NL5+ unit was equipped with an sCMOS camera (Teledyne Photometrix BSI express), providing a large Field of View of 18.8 mm (diagonal). Laser excitation from an Oxxius L4Cc laser diode combiner (containing a 638 nm laser) was coupled to the NL5+ module via an optical fiber. All experiments were conducted using laser power 7%, and a 60x air objective (Nikon, NA 0.95).

**Software and connections.** All components were controlled via Python. Specifically, pycromanager interacted with Micro-Manager

(version 2.0.3) to control laser powers, Z-stacks, and XY positioning. Additionally, the zaber_motion library was connected to the Zaber Launcher (version 2024.11.14) to control the autofocus device.

**Automated autofocus adjustments.** The autofocus loop involved several steps. To start, the objective was initially directed to a preset Z-position, aligning the autofocus laser within range for the first autofocus attempt. The autofocus was then triggered, aligning the objective with the bottom of the imaging plate. This in-focus focal height was recorded and serves as a reference for the next autofocus loop. After acquisition (see below), the autofocus routine subsequently started each new loop 10 μm below the previously recorded focal plane, searching upward to locate the plate bottom.

**XY Positioning and Image Acquisition.** The 96-well microscopy plate was mapped into 2 × 2 grids (550 μm spacing), creating technical replicates within each well. A well-specific event list was created, associating each well with the correct sample barcodes, coordinates, grid locations, and channel information. The scanning algorithm employed a snake pattern, optimizing acquisition time by minimizing travel distance and positional drift across the microscopy plate. Following autofocus, Z-stacks were captured as height additions on top of the recorded autofocus height, using dynamic spacing: a fine 0.5 μm step for the first 5 μm, increasing to 1.0 μm for the next 5 μm, then 2.5 μm for the following 5 μm, and finally 5.0 μm for deeper layers, spanning a total of 50 μm of Z-depth per position. Acquisitions were performed at 5 frames per second.

**Verification of imaging completion.** To monitor imaging progress, a continuous background process compared the number of saved image slices to the expected slice count based on the number of imaging events (i.e., focal planes across wells). Once the saved slice count matched the target, the acquisition was deemed complete, and MM and associated processes were automatically closed.

**Automated image analysis and classification.** Each acquired micrograph underwent automated analysis to extract sample classifications and particle features. Particles were detected using the scikit-image Python module. Yen thresholding was applied to create a binary mask, which was used to detect particles above 500 pixels ($5.87 \, \mu m^2$). The extracted particle properties (e.g., X/Y position, area, mean intensity) were saved for each micrograph. Results were then grouped by grid position and sorted by Z-index. For each particle, the slice with the largest detected area was selected as the representative view, which was used for the property mappings performed in this study. Wells were classified based on particle count and distribution, with 12 or more particles across at least three grid positions indicating "Phase Separation" and fewer particles marking "No Phase Separation".

## Machine learning and computation
**Design of the parameters space.** The initial dataset (i.e., cycle 0) for any given system formulation was created by computing a regular D-dimensional grid of points (with D being the number of variables to be considered), where each independent component of the formulation accounts for a dimension. Two of the dimensions were always assigned to the concentration of the two oppositely charged polymers, poly-L-(lysine) and poly-L-(aspartic acid) respectively. Additional dimensions could be added to account for other behaviors. The response variable was represented by an integer that mapped the recorded phase to either coacervate or not. In all our experiments we restrained our formulations to study the coacervation phenomena of two oppositely charged polymers as a function of the two polymer concentrations and the salt concentration. Additionally in the current work, we only focused on 2-D and 3-Dimensional datasets. This means that in the former case (2-D) the salt concentration is fixed and kept

constant, while in the latter case (3-D) it is allowed to change. The range for the polymer concentrations was constrained to be the same for all the experiments, regardless the polymer identity, and it was chosen to be a regularly spaced interval starting from concentration of 0.1 mM to 8.1 mM, with steps of 0.1 mM, giving a total of 81 concentrations values (end points included). Similarly, the range for the variation of the salt concentration was chosen to vary from 50 mM to 2075 mM with steps of 25 mM, giving a total of 81 values. All the ranges were chosen accordingly to the accuracy of the machines used to formulate the solutions. Finally, the dataset was created by filling a 2-D or 3-D regular grid with the values of the variable under investigation, creating a total of 6561 ($81 \times 81$) points for the 2-D case, and 531441 ($81 \times 81 \times 81$) for the 3-D case. In all the experiments the response variable was set to -1, the undefined default value, for all the points of the grid.

**Selection of new points.** Starting from cycle 0, and for each cycle, a subset of points $n$ (*i.e.*, new formulations) was sampled from the available pool of points $N$. The chosen sampling techniques followed the rules of Farthest Point Sampling (FPS)[47]. FPS is a sampling technique used to select a subset of points that are maximally spread out from each other within a given dataset. The goal is to retain points that represent the diversity of the data distribution by maximizing the minimum distance between selected points. Given a starting dataset $X = \{x_1, x_2, \ldots, x_N\}$ of $N$ points, a first random point $r_1 \in X$ was selected and added to the set of sampled points $S = \{r_1\}$. For each remaining point $x \in X \setminus S$ the minimum distance to any point in $S$ was computed:

$$d(x) = \min_{r \in S} ||x - r|| \tag{1}$$

Then, the point $x_i$ with the largest $d(x_i)$ was selected and added to the set of sampled points (*i.e.*, the point farthest from the currently sampled points). This selection was repeated until the desired number $n$ of points was reached. The result is a subset $S \subset X$ of $n$ points that were distributed in such a way that they maintain maximal separation, thereby capturing the structure of the original dataset more effectively than random sampling in cases where spread was important.

**Phase diagram (PD) prediction.** At each cycle $N$, a phase diagram was predicted using the data that has been experimentally tested in cycle $N - 1$. In the case of $N = 0$, no previous tested data was available, the prediction was skipped, and the FPS selected points were fed to the experimental validation pipeline, where their phase is recorded. For all $N \geq 1$, all the points assigned to the sampled set $S$, after experimental validation, would be used as the ground truth for a Gaussian Process Classifier (GPC)[46] model, that is going to predict the phase distributions over the entire input space. The GPC models the probability distribution over classes (e.g., the phases) by defining a latent function $f : \mathbb{R}^d \to \mathbb{R}^K$ that associates each input $x \in \mathbb{R}^d$, contained in the input space, with a set of probabilities $p(y = c|x, S)$, where $c \in \{1, \ldots, K\}$ represents the class labels. The training step involved using the subset $S$ to learn the posterior distribution of $f$, which, in turn, yielded a probabilistic model capable of assigning any points $x_i \in X$ to the probability of belonging to a specific class.

In our case, for each point in the input space the GPC would output a probability vector defined as follows:

$$\boldsymbol{p}_i = \left[ p(y = 1|x_i, S), p(y = 2|x_i, S) \right] \tag{2}$$

where each component represents the probability of $x_i$ belonging to either the "non-aggregate" ($y = 1$) or "coacervate" ($y = 2$) class. Obviously, given that $\boldsymbol{p}_i$ is a probability vector, it holds that the value for the sum of the individual contribution in Eq. 2 needed to sum up to 1. Thus, the GPC trained on the set of all the sampled and tested points could be used to provide a probabilistic prediction over the entire dataset, simply defined concatenating the individual vectors (Eq. 2) for

all the points contained in $X$,

$$\boldsymbol{P} = [\boldsymbol{p}_1, \ldots, \boldsymbol{p}_i, \ldots, \boldsymbol{p}_N] \tag{3}$$

Equation( 3) enabled inference about phase membership across all points, essentially representing the phase diagram.

The GPC algorithm used in our work was defined using a Radial Basis Function (RBF) kernel with length scale 1.0, multiplied by a constant kernel with default value of 1.0. In each application of the prediction algorithm, we allowed for an automatic internal optimization step by setting the parameters `n_restarts_optimizer` to 5, and the `max_iter_predict` to 150 (more information can be found on the original GPC Scikit-Learn documentation page[80]).

**Uncertainty estimation.** At each cycle, to estimate the uncertainty in the phase diagram predictions, we computed the information entropy for each point's probability vector $\boldsymbol{p}_i$ (Eq. 2). The uncertainty was computed as the (information) Entropy $H(x_i)$, and for $x_i$ could be computed as follow:

$$H(x_i) = - \sum_{c=1}^{K} p(y = c|x_i, S) \log p(y = c|x_i, S) \tag{4}$$

Higher entropy values indicate greater uncertainty, providing an uncertainty measure for each point in the phase diagram that is representative of the prediction's confidence level. The entropy range of values is bounded, and it depends on the number of independent classes $K$. In all our cases, $K = 2$, leading to a range of values that goes from $H = 0$, if either of the two classes was known for certain, *i.e.* $\boldsymbol{p}_i = [1.0, 0.0]$, to $H = 0.69$, if both of the two classes were most uncertain, *i.e.* $\boldsymbol{p}_i = [0.5, 0.5]$.

**Highest uncertainty landscape and exploration.** The values of $H(x_i)$ gave direct access to the so-called *uncertainty (phase) landscape* which represented, per cycle, which areas of the design space were most (un)certain. This information was then exploited to select a subset of points $X' \subset X$ that exhibited maximal entropy, within a set range of entropy values:

$$X' = \{x_i \in X | h \leq H(x_i) \leq H_{\max}\} \tag{5}$$

In Eq.( 4) the upper-bound limit, $H_{\max}$, represented the maximum value of entropy, defined as:

$$H_{\max} = - \sum_{c=1}^{K} \frac{1}{K} \log \frac{1}{K} = \log K \tag{6}$$

which for $K = 2$ it takes the value of $H_{\max} = \log 2 \approx 0.69$. The lower-bound limit can be freely chosen, and in our cases was set it to $h = 0.60$, effectively selecting only the highest uncertainty regions.

The points contained in $X'$ would then be used as the new search space for the FPS algorithm, sampling new suitable points for refining the prediction of the phase diagram. In the context of active learning, this was often referred to as the *exploration* phase of the cycle, where new points were selected trying to maximize the exploration, lowering the overall uncertainty of the predictive algorithm.

**Accuracy measurement.** To assess the accuracy of our classification model in a way that accounts for class imbalance, we used a balanced accuracy metric[59]. At each cycle a set of labels $Y^{(t)} = \{y_i^{(t)}\}$ was computed for each point in our dataset from the global vector of probabilities (Eq.( 3)). Balanced accuracy was defined as the average of the sensitivity for each class. Given the fact that we are dealing with a binary classification problem we can consider the 'coacervate' class as the 'positive' outcome and the 'non-aggregate' class as the 'negative' outcome.

Then, the balanced accuracy was defined as:

$$\text{Balanced Accuracy} = \frac{1}{2}\left(\frac{TP}{TP+FN} + \frac{TN}{TN+FP}\right) \quad (7)$$

In Eq. 7, $T_P$ and $T_N$ refer to the true-positive and true-negative predicted labels, while $F_P$ and $F_N$ refer to the false-positive and false-negative predicted labels.

**Convergence measurements.** To monitor the convergence of the model across cycles and/or experiment replicas, we tracked changes in the phase diagram, represented by the concatenated probability vector $\boldsymbol{P}^{(t)}$ that is outputted from the GPC prediction (Eq.( 3)). The superscript ($t$) indicates a cycle specific output of the probability vector. The convergence, in terms of Jensen-Shannon divergence (JSD)[61], could be computed in two main directions: across cycles and across replicas of experiments. The former required two different probability vectors, which belonged to two consecutive cycles of the same experiment, $\boldsymbol{P}^{(t)}$ and $\boldsymbol{Q}^{(t+1)}$, and it was defined as:

$$\text{JSD}\left(\boldsymbol{P}^{(t)}||\boldsymbol{Q}^{(t+1)}\right) = \frac{1}{2}\text{KL}\left(\boldsymbol{P}^{(t)}||\boldsymbol{M}^{(t,t+1)}\right) + \frac{1}{2}\text{KL}\left(\boldsymbol{Q}^{(t+1)}||\boldsymbol{M}^{(t,t+1)}\right) \quad (8)$$

where $\boldsymbol{M}^{(t,t+1)} = \frac{1}{2}\left(\boldsymbol{P}^{(t)} + \boldsymbol{Q}^{(t+1)}\right)$ represents the midpoint distribution. Each term on the right-hand side of Eq.( 8) represents the Kullback-Leibler divergence between one of the distributions and the midpoint. By computing the JSD between $\boldsymbol{P}^{(t)}$ and $\boldsymbol{Q}^{(t+1)}$ over successive iterations of the AL algorithm, we obtained a measure of convergence, with decreasing JSD values indicating stabilization of the model prediction across cycles. To compute the JSD across experiment replicas we could average the individual JSD measurements (Eq.( 8)) as follow,

$$\text{JSD}_{\text{replica}e}^{(t,t+1)} = \text{JSD}\left(\boldsymbol{P}_{\text{replica}e}^{(t)}||\boldsymbol{Q}_{\text{replica}e}^{(t+1)}\right) \quad (9a)$$

$$\overline{\text{JSD}}^{(t,t+1)} = \frac{1}{E}\sum_{e=1}^{E}\text{JSD}_{\text{replica}e}^{(t,t+1)} \quad (9b)$$

The average JSD value represented the convergence trend across multiple experimental replicas, which allowed to qualitatively account for the experimental variability.

**Overfitting tests and analysis.** To check for possible overfitting during the AL cycles we conducted some targeted robustness tests (Supplementary Fig. 50). First, we performed a parameter stress-test aimed at challenging the prediction ability of the trained model. We masked an increasing number of points from the screened pool and retrained the model (Supplementary Fig. 50B). The overall prediction remains almost unchanged, and even in the most extreme case, the overall prediction follows the expected result showcasing how the model is drawing the prediction in a smooth fashion, avoiding overfitting. Secondly, to test overfitting in a more systematic way, we employed the "y-scrambling" technique. In this approach, the target response values (the phase labels) of the training set are randomly shuffled, breaking any true relationship between input features and the target (Supplementary Fig. 50C). On average, the "scrambled" balanced accuracy is centered around 0.5 or lower, indicating that our approach indeed picks up relevant patterns in the underlying data.

**Downs-sampling search.** To test the performances of our pipeline with a coarser spaced grid we set up two different versions of the 2D and 3D search spaces. We systematically evaluated coarser sampling grids by selecting every third or fourth point. This yielded 9- to 15-fold reductions in the number of conditions in 2D (e.g., 6561 → 729 or 441)

and 27- to 53-fold reductions in 3D (e.g., 531,441 → 19,683 or 9,261). These new search spaces were used to run additional in-silico experiments comparing their performances to the default grid.

**Software and implementation.** All code regarding active machine learning was written in Python 3.12. The Python packages scikit-learn (v.1.5.0) was used for the implementation of the Gaussian Process Classifier and the calculation of the balanced accuracy. SciPy (v.1.13.1) was used for the computation of the information entropy. Pandas (v.2.2.1) was used to handle the datasets. All the other operations (*e.g.*, design space creation, farthest point sampling, and convergence calculation) were carried out with custom scripts using NumPy (v.<2.0.0). For data visualization, matplotlib (v.3.8.4) and plotly (v.5.9.0) were used in combination with Adobe Illustrator.

### Reporting summary
Further information on research design is available in the Nature Portfolio Reporting Summary linked to this article.

## Data availability
All data supporting this study are available in the article, the Supplementary Information, and the Source Data file (SourceData.xlsx). Raw and processed datasets from the active machine learning cycles, including those used to generate the manuscript and Supplementary Figs., are available on GitHub [https://github.com/molML/activeML-navigation-of-condensate-phases] and Zenodo [https://doi.org/10.5281/zenodo.17223126], together with instructions on how to replicate the figures. The complete confocal microscopy image dataset is too large for deposition in a public repository but is archived locally and can be made available by the corresponding authors upon request. Source data are provided with this paper.

## Code availability
The Python code to replicate and extend our active machine learning framework is openly accessible on GitHub at [https://github.com/molML/activeML-navigation-of-condensate-phases]. The code at the time of publishing is available at [https://doi.org/10.5281/zenodo.17223126].

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

## Acknowledgements

This work was supported by the National Growth Fund "Big Chemistry" funded by the Dutch Ministry of Education, Culture and Science (grant number 1420578 to J.C.M.H., F.G., L.B.). We gratefully acknowledge the Institute for Complex Molecular Systems (ICMS) for providing laboratory facilities. Special thanks to the Chemical Technology IT department, particularly Tom van Teeffelen and Frank Malipaard, for their expert advice and assistance in communication networks. We also extend our gratitude to Cristina Izquierdo Lozano for her support in data management and for the insightful discussions that enriched this work.

## Author contributions

Y.H.A.L., W.H., A.G., J.L.J.D., J.C.M.H., F.G., L.B. designed the automation pipeline. Y.H.A.L., W.H., A.G., and J.L.J.D. developed the automation pipeline. Y.H.A.L., W.H., A.G., A.R-A., N.A.E., J.C.M.H., F.G., L.B. designed the experiments. Y.H.A.L., W.H., A.G., A.R-A, and N.A.E. performed the experiments. Y.H.A.L., W.H., A.G., A.R-A., N.A.E., J.C.M.H., F.G., L.B. analyzed the data. Y.H.A.L., W.H., A.G., J.C.M.H., F.G., and L.B. wrote the manuscript. J.C.M.H., F.G., and L.B. supervised the study. All authors reviewed the manuscript.

## Competing interests

The authors declare no competing interests.
