## [Transparent Peer Review file · Nature Communications]

Automated navigation of condensate phase behavior with active machine learning

Corresponding Author: Professor Luc Brunsveld

Version 0:

Reviewer comments:

Reviewer #1

(Remarks to the Author)

The authors developed an automated condensate phase navigation platform with active machine learning algorithm. This work is very interesting and I really like it. The authors might need to do more to make this work more useful.

1. The searching space seems not very large. only 81×81 for 2D and $81 \times 81 \times 81$ for 3D. For such a space, active learning may not fully bring into play. I suggest a test on larger space, for example, 4 parameters. This will give more confidence to the work.
2. The physical meaning of condensate phase behavior after the navigation is suggested to be explored and discussed. That is, I hope this work help to tell us something why it happened.
3. Minor point: Some Refs, for example, 49 and 60 are not superscripted.

(Remarks on code availability)

I can run the code alone as it is based on the automatic machine.

Reviewer #2

(Remarks to the Author)

This manuscript describes an automated approach to deriving phase diagrams for condensates. The approach is elegant, fully automated, and, thanks to active machine learning, very efficient. While I am not an expert in automation, having done a little in my laboratory, I recognize the value of a paper like this. As far as I understand, the approach is correct, novel, elegant, and important.

My expertise would be more on the thermodynamic meaning of the data presented, and with this part of the manuscript I do have problems. Condensates are complex, and trivialization of their behavior can be misleading. When two liquid phases separate they should do so in bulk (the lower-density one occupies the top part, the other the lower). It does happen that droplets can be found, and this can be because of surface energy considerations or local thermodynamics minima, or simply kinetic traps. In the manuscript, the author describe the complex behavior they observe but do not discuss whether what they see is a kinetic trap or a thermodynamic effect, and in both cases, they fail to discuss how to properly represent this in a phase diagram. For example, some observations may simply indicate that the system is far from equilibrium but trapped in a local energy minimum, in such case, it is questionable if it belongs in a phase diagram or whether the lines representing it should be dotted. Finally, when I look at the phase diagrams I see re-entrant phases, something that I cannot physically explain and that is not discussed in the manuscript, this makes me think that indeed the phase diagram is incorrect (probably because of kinetic traps, I do not doubt the validity of the experimental method).

Given these considerations, I am truly uncertain about this manuscript, as the experimental part is elegant, the results described and the discussion is really far from being acceptable.

For clarity, for the re-entrant phases I am referring to Figure 5 G-H-I; I am currently thinking that they are an over fitting from the ML, but I am not an expert.

(Remarks on code availability)

I am not an expert, I have not reviewed the code, it would have made no sense.

Reviewer #3

(Remarks to the Author)

Summary

Leurs and colleagues present an automated analysis of liquid condensate phase behaviour. Pipetting robotics and confocal microscopy supplemented with machine learning map and predict phase diagrams in an automated iterative fashion. An uncertainty measure is invoked to prioritize experimental effort to systematically underexplored places in phase space. While this reviewer appreciates the numerous technical achievements, the shallow theoretical analysis of the obtained data is underwhelming. Major revisions are hence suggested.

Weaknesses:

1) The advanced automation implemented by the authors achieved an abundance of information-rich data, including the quantification of key condensate properties such as particle size, count, and volume fraction. Such quantitatively data invites to careful theoretical analysis. How could the authors for instances extract specific thermodynamic information of the investigated phase transitions? For instances, how can solute chemical potentials be extracted? How can coexistence tie-lines be inferred? How can binodals and spinodals be quantified correctly? Unfortunately, rather than pursuing any of these questions the authors seem keen to completely avoiding the subject. Numerous theoretical frameworks exist, beyond those already cited by the authors, we like to suggest following perhaps more overlooked manuscripts as a possible entry: <https://doi.org/10.1063/1.474547> , <https://pubs.acs.org/doi/pdf/10.1021/acs.jpcc.0c00402> , and <https://pubs.acs.org/doi/full/10.1021/acs.chemrev.2c00814>

2) The authors state they were able to analyse „condensate properties beyond phase boundaries“. This seems however limited to computing the average area covered by condensates and their volume fraction. Obtaining these data accurately at scale has technical implications, however it remained elusive how these values can critically inform on the systems phase separation. To our understanding important aspects of condensates are material properties likes viscosity, diffusivity of proteins, partitioning coefficient or Gibbs free energy (see <https://doi.org/10.1038/s41587-019-0341-6> and <https://doi.org/10.1038/s41586-020-2256-2>). What specific value could the authors extract from these analysed covered area & volume fraction?

3) Two peptides of opposing charges where investigated. While a good model to demonstrate the basic utility of the chosen approach, their over-simplistic phase behaviour fails to address typical problems of assessing phase behaviour in more complex compounds such those inferred from high viscosity or aggregate formation (i.e. <http://dx.doi.org/10.1016/j.molcel.2015.08.018> , Fig 1B). Scikit may score non-roundish appearance of highly viscous condensates by an ML algorithm, but how did the authors suggest to navigate aggregation or glass transitions? Can the algorithm differentiate them from condensates? How does it handle aggregates below the 500 px threshold?

4) Overall sample consumption appears unforgivingly high: 150 μ L per well and 40-250 samples for a phase diagram resulting in 6 to 37 mL of total sample. Protein purification can be elaborate. Thus, a reasonable sample consumption is crucial for the system to be adopted, yet along recognized in the field. Protein amounts consumed per phase diagram should be reported for all diagrams.

Also, are the authors aware of manuscripts where such amounts of material were available to investigate macromolecular phase separation. Concurrently, how would a 10, 100, or 1000 fold reduction in sample consumption be achievable? Why did the authors not explore these?

Additional remarks:

- BSA is used to passivate the surface of the well-plates. However, addition of BSA can also work as a crowding agent and thus influence phase separation behaviour. We would like to see a control experiment that this is not the case for the implemented protocol.
- The authors used a custom tip with a „touch“ function. To allow for accurate reproducibility all essential methodological details must be communicated clearly. We thus like would ask the authors to provide an explanation of the exact tip function in the method section.

(Remarks on code availability)

Reviewer #4

(Remarks to the Author)

(Remarks on code availability)

Version 1:

Reviewer comments:

Reviewer #1

(Remarks to the Author)

Though the authors did not fully answer my questions, I can accept their responses in current form.

(Remarks on code availability)

The code is okay and it's really time-consuming to go through all the code.

Reviewer #2

(Remarks to the Author)

I am totally satisfied with the replies to my comments.

(Remarks on code availability)

Reviewer #5

(Remarks to the Author)

The expansion of experiments and additional clarifications satisfy all of the major concerns that were raised. The additional validations and inclusion of data make for a stronger, clearer manuscript.

One lingering subject comes up from reviewing the authors' responses, though it does not require changes. If the system has the advantage of detecting larger coacervate droplets, but misses smaller, but significant quantities of smaller droplets, what is the significance or advantage of detecting larger droplets beyond extending the range of measurements above an NTA? Are larger droplets more physiologically relevant, etc. This brings in the definition of what a LLPS phase diagram is, and if it is determined from first principles then size-limited measurements are not likely to yield results that match theoretical predictions.

For the purpose of this publication, it is unnecessary to resolve this question, and inclusion of the orthogonal data allows an informed reader to interpret the experiments as presented.

The addition of simulations to validate that the methodology for more complex phase boundaries is very encouraging and a great addition that demonstrates the potential beyond the repetitive (but meaningful) curves the manuscript focuses on. I look forward to the extension of this approach to other materials and believe it could be a phenomenal tool for first advancing first-principle analysis, when phase boundaries are predicted and the automated systems can be used to validate or refute the predicted results or train a reasoning model. Tuning this system to investigate edge cases or identify the limits of theories could be as powerful as the ability to classify a large number of materials to establish a database.

I strongly recommend publication of this revised manuscript.

Additionally, after reading the combined reviews from reviewers 3 & 4 and responses from the authors again, I would say that the answers provided are sufficient.

The discussion around comment 2 was probably the weakest response by the authors. From the reviewers' closing comment:

"To our understanding, important aspects of condensates are material properties like viscosity, diffusivity of proteins, partitioning coefficient or Gibbs free energy (see <https://doi.org/10.1038/s41587-019-0341-6> and <https://doi.org/10.1038/s41586-020-2256-2>). What specific value could the authors extract from these analysed covered area & volume fraction?"

The authors explain why they can't use three established techniques, then mention the partition coefficient provides thermodynamic parameters, but they don't demonstrate that or provide confirmation that their partition coefficients match any of the three established methods. Since this is unproven in a rigorous manner, they might want to lighten the claim. It seems like they are explaining why their method isn't authoritative but serves as a strong practical surrogate for better defined methods. That could be followed up on without high throughput.

Generally, their response to this comment is one of clarifying what they are claiming, but not necessarily just saying, no it isn't proven effective for that at the present time.

(Remarks on code availability)

The code was clear to read and available without issues. I did not execute the code, but believe I would be able to from what was provided. The logic of the code was also clear.

Reviewer #1

The authors developed an automated condensate phase navigation platform with active machine learning algorithm. This work is very interesting and I really like it. The authors might need to do more to make this work more useful.

Response:

We sincerely thank Reviewer #1 for their positive evaluation and encouraging feedback. We appreciate their recognition of our automated condensate phase navigation platform and its integration of active machine learning. The reviewer's suggestions have been valuable, and we have addressed them to improve the clarity and practical utility of our approach. We share their interest in the broader impact of this work, which we see as a step toward a general, adaptable framework for autonomous exploration of high-dimensional phase spaces, particularly where exhaustive sampling is infeasible.

Comment 1:

The searching space seems not very large. only 81×81 for 2D and $81 \times 81 \times 81$ for 3D. For such a space, active learning may not fully bring into play. I suggest a test on larger space, for example, 4 parameters. This will give more confidence to the work.

Response:

While our 2D (6,561 points) and 3D (531,441 points) search spaces are finite, they are already far too large to explore manually – highlighting the practical need for efficient sampling strategies like active learning (AL). Rather than expanding our experimental space to four parameters, we rigorously assessed our current AL framework by benchmarking it against standard alternatives.

To this end, we conducted in-silico comparisons using the full ground-truth data available for our 2D and 3D systems (see figure below, now added as Supplementary Figure S7). We evaluated three strategies: AL, random sampling, and regular grid sampling. In each round, new data points were selected and used to retrain a Gaussian Process Classifier. Even in 2D, AL consistently achieved higher balanced accuracy with fewer samples and lower variance, demonstrating its clear advantage.

Supplementary Figure S7: Comparison of Active Learning (AL) and non-Machine Learning (ML) sampling methods for phase diagram screening. A) Results for the 2D system (poly-*L*-(lysine)₁₀₀ / poly-*L*-(aspartic acid)₂₀₀; 6561 points), showing the balanced accuracy between the predicted phase diagrams and the collected ground truth for three sampling strategies: in-silico AL (solid purple line), random sampling (dashed brown line), and regular grid sampling (dash-dotted dark green line). In each iteration, selected data points were used to retrain a Gaussian Process Classifier, with AL following the same iterative procedure as in the experimental workflow. Replicates: random ($n = 5$), AL ($n = 3$), grid (deterministic). **B)** Results for the 3D system (poly-*L*-(lysine)₁₀₀ / poly-*L*-(aspartic acid)₂₀₀ / NaCl; 531,441 points), using the same evaluation as (A). Shaded regions indicate the standard deviation computed over multiple replicates ($n=5$ random sampling, $n=3$ for AL; grid sampling is deterministic).

Expanding the parameter space to four dimensions or more is an exciting and meaningful direction for future work. Such a space would vastly increase the experimental complexity and time (easily resulting in tens of millions of possible combinations) and further amplify the need for data-efficient exploration strategies. While this lies beyond the current manuscript's scope, we anticipate that scalable models such as Bayesian Neural Networks could complement our framework in navigating such a high-dimensional phase space.^{1,2}

Comment 2:

The physical meaning of condensate phase behavior after the navigation is suggested to be explored and discussed. That is, I hope this work helps to tell us something about why it happened.

Response:

The observed condensate behavior is governed by the physical principles of associative liquid–liquid phase separation (LLPS), driven by the minimization of free energy. This occurs when the energetic gains from transient, weak interactions, such as electrostatic attractions between oppositely charged components, combined with the entropic benefit of releasing bound solvent molecules, together outweigh the entropic penalty associated with reduced conformational freedom.^{3,4}

These molecular principles are reflected in the experimental phase diagrams (Figure 2B), which show strong phase separation near stoichiometric ratios of the two polyelectrolytes and above a certain concentration threshold. However, further increasing the concentration of one of the polyelectrolytes disrupts phase separation, likely due to charge inversion and electrostatic shielding, which prevent the formation of an extended polymer network.⁵

We also examined the influence of critical parameters such as salt concentration (Figure 6), where both low and high salt conditions inhibit LLPS. This aligns with previous studies and illustrates the complex interplay of ionic strength and molecular interactions in phase behavior.⁶ These results underscore the capability of our platform to systematically explore the multi-parameter space of phase separation and map complex molecular interactions in micron-scale systems.

We appreciate the reviewer's suggestion and have revised the manuscript to deepen the discussion of the physical and molecular mechanisms underlying the observed behaviors:

“Notably, the resulting shape of the phase diagram is consistent with the physical principles underlying condensate formation. In associative LLPS, droplet formation is driven by multivalent electrostatic interactions and the release of bound solvent molecules, which together must outweigh the entropic cost of reduced chain flexibility^{54,55}. These conditions are optimally met near stoichiometric charge ratios, where attractions between the polypeptides are maximized. When one component is in excess or depleted, the resulting charge imbalance and electrostatic screening hinder the formation of an extended interaction network, thereby suppressing phase separation as reflected in the mapped binodal.⁵⁶”

Comment 3:

Minor point: Some Refs, for example, 49 and 60 are not superscripted.

Response:

This formatting issue has been corrected in the revised manuscript. All references have now been consistently formatted and properly superscripted.

Comment on Code Availability:

I can run the code alone as it is based on the automatic machine.

Response:

We thank the reviewer for their positive comment. The provided code and in-silico setup were specifically designed to allow users to independently run active learning cycles using the defined phase diagrams, enabling full reproducibility and exploration without the need for laboratory automation.

Reviewer #2

This manuscript describes an automated approach to deriving phase diagrams for condensates. The approach is elegant, fully automated, and, thanks to active machine learning, very efficient. While I am not an expert in automation, having done a little in my laboratory, I recognize the value of a paper like this. As far as I understand, the approach is correct, novel, elegant, and important.

Response:

We sincerely thank Reviewer #2 for their thoughtful and generous comments. We appreciate their recognition of the novelty, efficiency, and relevance of our automated approach. Their appreciation of automation and active machine learning is especially encouraging, as these elements are central to both our methodology and the broader goals of our work.

Comment 1:

In the manuscript, the author describe the complex behavior they observe but do not discuss whether what they see is a kinetic trap or a thermodynamic effect, and in both cases, they fail to discuss how to properly represent this in a phase diagram.

Response:

Phase separation can indeed arise from thermodynamic instability (where demixing lowers the free energy) or from kinetic traps (such as arrested phase separation like gelation). In this work, we mapped phase behavior by identifying an operational boundary that separates samples with observable coacervates from those without. For accurate interpretation, it is indeed critical to consider whether this boundary corresponds to the binodal, the spinodal, or neither, and whether the resulting diagram reflects equilibrium behavior or is better described as a state diagram.

To address this point, we have now also constructed phase diagrams of poly-*L*-(lysine)₁₀₀ and poly-*L*-(aspartic acid)₂₀₀ by imaging samples after defined incubation periods of 5, 15, 30, and 60 minutes (see figure below, now added as Supplementary Figure S22). Each diagram was generated as a separate experiment, guided by machine learning, to adequately probe the evolving phase boundaries. Notably, the 15-minute time point corresponds to the standard incubation period used throughout the main manuscript. We observed that the phase boundary shifts noticeably between 5 and 15 minutes, consistent with ongoing droplet formation, settling, and coalescence during the early stages. However, from 15 minutes onward, the boundary remains stable, with no appreciable differences observed between the 15-, 30-, and 60-minute conditions. These results indicate that by 15 minutes, the system reaches its near-equilibrium phase-separated or non-phase-separated state, supporting our interpretation that the boundary observed at this time reflects a near-equilibrium representation of phase behavior rather than a transient or kinetically arrested state.

Supplementary Figure S22: Time-resolved phase diagram mapping of poly-L-(lysine)₁₀₀ / poly-L-(aspartic acid)₂₀₀ coacervates. Phase diagrams were constructed by imaging after incubation times of 5, 15, 30, and 60 minutes. Each condition was treated as a separate experiment. The operational phase boundaries are shown as dashed lines for each time point. Substantial shifts in the phase boundary are observed between 5 and 15 minutes, indicating ongoing droplet settling. Beyond 15 minutes, the phase boundary remains stable, suggesting that the system has reached a near-equilibrium state under the experimental conditions. Polypeptide concentrations are reported in monomer concentrations.

To further support this conclusion, we performed time-resolved turbidity and dynamic light scattering (DLS) measurements (figure below, now added as Supplementary Figure S25). These orthogonal readouts show that bulk demixing plateaus around the 15-minute mark, validating its use as a consistent and practical time point for binodal measurement. We therefore consider the diagrams in the manuscript to be phase diagrams that approximate the binodal under near-equilibrium conditions.

Notably, while binary classification is stable by this time, coalescence and sedimentation continue to evolve (see our response to Reviewer 5, Comment 4). We have now added Supplementary Figures S22 and S25 and have explicitly clarified the incubation time in the Methods section (see our response to Reviewer 2, Comment 2).

Supplementary Figure S25: Time-resolved measurements validating the 15-minute incubation time used in the automated pipeline. (A) Turbidity kinetics of poly-*L*-(lysine)₁₀₀ / poly-*L*-(aspartic acid)₂₀₀ formulations near the phase boundary. Poly-*L*-(lysine)₁₀₀ was fixed at 2.00 mM, while poly-*L*-(aspartic acid)₂₀₀ was varied from 0.1 to 1.00 mM. (B–D) Time correlation functions obtained by dynamic light scattering (DLS) for the same formulations, measured between 0 and 25 minutes post-mixing (timepoints indicated in the legend). The decay of the autocorrelation function indicates the diffusion behavior of the coacervates, which is linked to particle size via the Stokes-Einstein equation. These correlograms, particularly in panels C and D, reveal a progressive flattening that reflects changes in particle size distribution and droplet growth over time. This flattening plateaued around 15 minutes for each sample, supporting the use of the 15 minute-mark as a consistent and practical snapshot for coacervate quantification. Polypeptide concentrations are reported in monomer concentrations.

Comment 2:

For example, some observations may simply indicate that the system is far from equilibrium but trapped in a local energy minimum, in such case, it is questionable if it belongs in a phase diagram or whether the lines representing it should be dotted.

Response

We agree that distinguishing equilibrium states from kinetically trapped ones is essential when interpreting and representing phase behavior. To examine long-term stability, we monitored selected samples over extended periods of 1, 2, 3, and 21 days (see figure below, now added as Supplementary Figure S24). Coacervates remained clearly visible throughout this period on passivated surfaces, retained their ability to fuse, and showed no signs of irreversible aggregation or solidification. These observations provide strong support that the classified states are not transient, kinetically trapped artifacts.

Supplementary Figure S24: Time-lapse imaging of coacervates formed by poly-L-(lysine)₁₀₀ and poly-L-(aspartic acid)₂₀₀. Confocal micrographs show selected formulations with distinct coacervate morphologies alongside a non-phase-separating control panel, imaged at 1 day, 2 days, 3 days, and 21 days after sample preparation. Samples were stored at room temperature in a sealed microplate and re-imaged over time by repeatedly moving the plate in and out of the microscope. A slight drift in field of view is visible at later time points due to repositioning. Across all coacervation conditions, droplets remained clearly visible and retained their morphology over three weeks, with no signs of structural maturation or solidification. Colored boxes highlight selected droplet fusion events over time. For transparency, micrographs were contrast-enhanced to allow visualization of faint background signals in the non-coacervating samples and to confirm the absence of phase separation. Scale bars: 20 μm .

Combined with the time-resolved experiments presented in Supplementary Figures S21 and S24 (Comment 1, Reviewer 2), we are confident that the operational boundary defined in the manuscript reflects the binodal under near-equilibrium conditions. We do note that the incubation time required to reach such a state can be highly system dependent. The polypeptides used in our study do not undergo structural maturation, such as β -sheet or fibril formation, which is known to occur in protein systems like FUS or tau and can alter material properties over time.⁷⁻¹⁰ In light of this, we have added a clarification in the Methods section regarding incubation time selection.

“Notably, our property measurements represent standardized snapshot observations taken after a 15-minute incubation period of a dynamic, continuously evolving system. While the incubation time can be adjusted to suit the user’s objective or extended to enable kinetic measurements (Supplementary Figure S22-24), we selected 15 minutes as a practical and reproducible readout, based on DLS and turbidity data showing that key features for identifying phase separation begin to stabilize around this time (Supplementary Figure S25).”

Comment 3:

Finally, when I look at the phase diagrams I see re-entrant phases, something that I cannot physically explain and that is not discussed in the manuscript, this makes me think that indeed

the phase diagram is incorrect (probably because of kinetic traps, I do not doubt the validity of the experimental method).

For clarity, for the re-entrant phases I am referring to Figure 5 G-H-I; I am currently thinking that they are an over fitting from the ML, but I am not an expert.

Response:

The "*dilute–condensed–dilute*" transitions observed upon increasing the concentration of one of the polyelectrolytes are indeed consistent with charge inversion effects in systems undergoing associative phase separation. Similar profiles have been reported in literature, for example in RNA–polycation and other oppositely charged macromolecular systems.^{5,6,11} These transitions arise from electrostatic imbalance at high concentration ratios and are well-grounded in the physics of complex coacervation.

If the comment refers to a *condensed–dilute–condensed* transition along a single thermodynamic axis, we would like to clarify that such behavior is not observed. In Figures 5G and 5H, the phase transitions are monotonic along both axes. In Figure 5I, a slight undulation near the lower phase boundary may resemble non-monotonicity, but this occurs over a narrow range and is unlikely to be physically meaningful.

To directly address the concern regarding possible overfitting of the phase boundaries, we conducted some targeted *in silico* robustness analyses (see figure below, now added as Supplementary Figure S50).

Supplementary Figure S50: Overfitting tests and analysis. A) Summary of the results for the 2D phase diagram exploration of poly-*L*-(lysine)₂₅₀ and poly-*L*-(aspartic acid)₂₀₀. The filled circles represent the screened points and are red when the recorded phase is not an aggregate or blue when the recorded phase is a coacervate. The fine grid in the background is the total 2D search space with superimposed the model prediction based on the collected experimental evidence. B) Stress-test of the model prediction by masking 8, 12, 16, or 20 of the screened points, represented by an 'X' in the figure. The masking points are chosen at random amongst the available ones. C) Y-scrambling test for each 2D experiments in main text Figure 5.

First, we performed a parameter stress-test aimed at challenging the prediction ability of the trained model. We masked an increasing number of points from the screened pool and retrained the model. Panel B showcases the outcome for a masking of 8, 12, 16, and 20 points. The overall prediction remains almost unchanged, and even in the most extreme case of masking 20 points, the overall prediction follows the expected result. This test showcases how the model is drawing the prediction in a smooth fashion, avoiding overfitting specific configurations.

Secondly, to systematically exclude overfitting, we employed the well-established “y-scrambling” technique. In this approach, the target response values (the phase labels) of the

training set are randomly shuffled, breaking any true relationship between input features and the target. The scrambled data is then used to train the ML model, and its predictive performance is compared against that of the original, unscrambled model. This technique helps assess whether the model is truly learning meaningful patterns or merely overfitting noise or spurious correlations within the limited labeled data (often the case in the low-data regimes of Active Learning). Panel C shows the results for 500 repetitions of the y-scrambling experiment for all the diagrams reported in Figure 5 in the main manuscript. On average, the “scrambled” balanced accuracy is centered around 0.5 or lower (random performance, or worse), indicating that our approach indeed picks up relevant patterns in the underlying data and is not overfitting.

Hence, the data presented do not support the presence of a convincing *condensed–dilute–condensed* regime. The features observed are consistent with known physical principles and are experimentally reproducible. We have now included Supplementary Figure S50 and added the following discussion to our revised manuscript methods section:

“Overfitting tests and analysis. To check for possible overfitting during the AL cycles we conducted some targeted robustness tests (Supplementary Figure S50). First, we performed a parameter stress-test aimed at challenging the prediction ability of the trained model. We masked an increasing number of points from the screened pool and retrained the model (Supplementary Figure S50B). The overall prediction remains almost unchanged, and even in the most extreme case, the overall prediction follows the expected result showcasing how the model is drawing the prediction in a smooth fashion, avoiding overfitting. Secondly, to test overfitting in a more systematic way, we employed the “y-scrambling” technique. In this approach, the target response values (the phase labels) of the training set are randomly shuffled, breaking any true relationship between input features and the target (Supplementary Figure S50C). On average, the “scrambled” balanced accuracy is centered around 0.5 or lower, indicating that our approach indeed picks up relevant patterns in the underlying data.”

Comment on Code Availability:

I am not an expert, I have not reviewed the code, it would have made no sense.

Reviewer #3&4:

Leurs and colleagues present an automated analysis of liquid condensate phase behaviour. Pipetting robotics and confocal microscopy supplemented with machine learning map and predict phase diagrams in an automated iterative fashion. An uncertainty measure is invoked to prioritize experimental effort to systematically underexplored places in phase space. While this reviewer appreciates the numerous technical achievements, the shallow theoretical analysis of the obtained data is underwhelming. Major revisions are hence suggested.

Response:

We thank the reviewers for recognizing the technical contributions of our work. We would like to clarify that this work was developed and submitted in the context of a special issue focused on laboratory automation. As such, the primary aim of the manuscript is to demonstrate and validate a modular, extensible platform for automated, data-driven exploration of phase behavior in complex supramolecular systems. The emphasis is therefore placed on the experimental pipeline, the iterative learning strategy, and the robustness of the automation framework rather than on an in-depth theoretical and thermodynamical analysis of the phase behavior observed.

That said, we agree that physical interpretation of the acquired data is an important aspect. In response to the reviewer's suggestion, we have performed several additional control experiments and expanded the theoretical framing of our results. While a comprehensive thermodynamic analysis remains beyond the primary scope of this automation-focused study, we now discuss key features in the observed phase diagrams in relation to established theoretical models and literature examples. These revisions aim to provide a more balanced integration of physical insight and experimental design. The specific additions and changes are detailed in our responses to the individual comments below.

Comment 1:

The advanced automation implemented by the authors achieved an abundance of information-rich data, including the quantification of key condensate properties such as particle size, count, and volume fraction. Such quantitatively data invites to careful theoretical analysis. How could the authors for instances extract specific thermodynamic information of the investigated phase transitions? For instances, how can solute chemical potentials be extracted? How can coexistence tie-lines be inferred? How can binodals and spinodals be quantified correctly? Unfortunately, rather than pursuing any of these questions, the authors seem keen to completely avoid the subject. Numerous theoretical frameworks exist, beyond those already cited by the authors, we like to suggest following perhaps more overlooked manuscripts as a possible entry: <https://doi.org/10.1063/1.474547>, <https://pubs.acs.org/doi/pdf/10.1021/acs.jpcc.0c00402>, and <https://pubs.acs.org/doi/full/10.1021/acs.chemrev.2c00814>

Response:

We appreciate the reviewers' many suggestions to further gain quantitative information from the application of the platform and have expanded our analysis below. We also invite the reviewers to read our response to Reviewer 2, where we present additional experiments confirming that the system operates near thermodynamic equilibrium under our standard conditions (Supplementary Figures S22-25). We agree that there is exciting potential to obtain further information about the system. Notably, there are some practical limitations to consider with respect to our experimental setup.

Firstly, the platform, spanning multiple pieces of equipment, is always operating at room temperature. Lacking the ability to vary temperature means we cannot determine the critical temperature, $X(T)$, and cannot decouple the enthalpic and entropic contribution to the free energies.

Secondly, we lack direct access to the “dense phase branch” of the binodal. As an initial attempt to estimate it (Figure R1), we applied the lever rule to construct tie lines using total (black) and dilute (blue) phase concentrations, together with measured dense-phase volume fractions from Main Text Figure 4 and the assumption of a tie line slope of +1. Since our measurements were taken near the phase boundary, the dense-phase volume fractions are below 0.2%, where small deviations in the measurements cause a very significant error when extrapolating using the lever rule. This is after already excluding datapoints in which the dense phase fraction is below 0.01%. Here, we have converted concentrations to volume fraction using a partial specific volume of 0.73 mL/g, a typical value for unfolded polypeptides. In several cases, the extrapolated dense-phase values exceeded a volume fraction of 1, confirming that this approach is unreliable under these conditions.

Figure R1: Extrapolation of dense-phase compositions near the binodal is unreliable due to amplified error. Shown are total (black), dilute (blue), and extrapolated dense-phase compositions (red) for samples with dense-phase fractions $\alpha \geq 0.01$, based on the data in Main Text Figure 4. Tie lines were calculated using the lever rule, assuming mass balance. However, since the dense-phase volume fractions are very low near the phase boundary (often $<0.2\%$), even small measurement errors lead to large inaccuracies in the extrapolated values. Several points exceed physically plausible volume fractions, demonstrating that this method is not reliable in this regime and supporting the need for direct measurement approaches as used in Supplementary Figure S26.

To address this, we aimed to directly measure the dense-phase concentrations of both components using dual-color confocal microscopy (see figure below, now added as Supplementary Figure S26). For this purpose, we covalently labeled the polymers with fluorescent dyes to generate sulfo-Cy5-labeled poly-L-(lysine)₁₀₀ and FITC-labeled poly-L-(aspartic acid)₂₀₀. Samples were imaged simultaneously in the 640 nm and 488 nm channels, using the Cy5 signal to segment the coacervates (panels A-C & F). This mask was then applied to both fluorescence channels to extract mean intensities within the dense phase (panels D &

G). Intensities were converted to local concentrations using carefully established calibration curves from homogeneous dye solutions of known concentration (panels E&H).

This approach enabled us to obtain estimates of the dense phase concentrations for both components. We measured $[\text{FITC}]_{\text{dense}} = 3.6 \pm 0.5 \mu\text{M}$ and $[\text{Cy5}]_{\text{dense}} = 1.3 \pm 0.2 \mu\text{M}$, corresponding to a 14.4 ± 2 -fold enrichment of the FITC-labeled poly-L-(aspartic acid) and a 5.2 ± 0.8 -fold enrichment of the Cy5-labeled poly-L-(lysine), relative to the initial input concentration of 250 nM. With respect to poly-L-(lysine) and poly-L-(aspartic acid), the tie-line should have a positive slope. This is consistent with the behavior expected for associative complex coacervation, where both oppositely charged components are enriched in the dense phase. The measured tie-line slope indeed reflects the underlying thermodynamics of the system and aligns with theoretical expectations for binary complex coacervates undergoing liquid–liquid phase separation.

In future work, we will extend our automation pipeline to directly measure the dilute phase concentration to derive partitioning coefficients in an automated manner. Notably, a final limitation of this platform is that we cannot directly measure partitioning for components which cannot be covalently fluorescently labelled, therefore partition coefficients for components like the buffer, Na^+ and Cl^- cannot be determined with our platform. Literature reports steep tie-lines and non-one partitioning coefficients for ionic species for various coacervate systems.¹² As such, we feel it is appropriate to report the dense phase measurements work as a measurement of a “projected tie line” or “partition coefficient”, rather than a “tie line”.

As the reviewers point out, we are conservative in extracting thermodynamic information from our measurements. This is a consideration with respect to the platform limitations and keeping the focus of the manuscript on the main advancement: Automatic and efficient generation of binodals unlike previously possible. Notably, these considerations and our measurements shown in Supplementary Figure S26 were previously not included in the manuscript. Given that readers may have similar questions as the reviewers, we have now included this and the references mentioned by the reviewers.

Supplementary Figure S26: Measurement of dense-phase dye concentrations using dual-color confocal imaging. (A) Schematic of the confocal imaging setup with dual-camera acquisition. A longpass filter splits emission signals, enabling simultaneous imaging of FITC (488 nm) and Cy5 (640 nm) channels. (B) Automated z-stack acquisition from coacervate samples, capturing condensates across multiple focal planes. (C) Example segmentation of the Cy5 channel, with detected condensates outlined in red. (D) Spatial map of all detected Cy5-positive particles across four grid positions and z-planes, color-coded by mean Cy5 fluorescence intensity. (E) Calibration curve relating Cy5 fluorescence intensity to known Cy5 concentrations ($R^2 = 0.998$), enabling conversion of mean intensity values to local dye concentrations. (F) Cy5 mask from panel (C) overlaid on the FITC channel to extract FITC signal from the same particles, shown with green outlines. (G) Spatial map showing the corresponding FITC intensity values for each segmented region. (H) Calibration curve for FITC fluorescence versus concentration ($R^2 = 0.9995$), enabling quantification of local FITC concentration.

Data shown correspond to a sample containing 6.1 mM poly-*L*-(lysine)₁₀₀ and 8.1 mM poly-*L*-(aspartic acid)₂₀₀ (monomer concentrations), each spiked with 250 nM of fluorescently labeled polypeptides. The resulting dense-phase concentrations were estimated to be $[\text{Cy5}]_{\text{dense}} = 1.3 \pm 0.2 \mu\text{M}$ and $[\text{FITC}]_{\text{dense}} = 3.6 \pm 0.5 \mu\text{M}$.

Comment 2:

The authors state they were able to analyse „condensate properties beyond phase boundaries“. This seems however limited to computing the average area covered by condensates and their volume fraction. Obtaining these data accurately at scale has technical implications, however it remained elusive how these values can critically inform on the systems phase separation. To our understanding important aspects of condensates are material properties like viscosity, diffusivity of proteins, partitioning coefficient or Gibbs free energy (see <https://doi.org/10.1038/s41587-019-0341-6> and <https://doi.org/10.1038/s41586-020->

2256-2). What specific value could the authors extract from these analysed covered area & volume fraction?

Response:

We agree with the reviewer on the importance of material properties such as viscosity, diffusivity, partitioning coefficients, and Gibbs free energy in understanding condensate behavior. These parameters offer direct thermodynamic and mechanistic insight, and have indeed advanced the field significantly, as exemplified by Bracha et al. and Riback et al., whose work we now cite in the revised manuscript.

That said, such metrics are inherently challenging to obtain in high-throughput workflows. Techniques like FRAP require targeted photobleaching and specialized automated microscopy routines; microrheology relies on embedded tracer particles that robustly partition into droplets regardless of the formulation conditions; and optical trapping setups require additional equipment that are not easily fully automated. Generally, these requirements limit throughput and scalability, making it difficult to extract such properties across hundreds of conditions in an iterative screen.

Our platform was specifically designed to provide an intermediate solution: it enables scalable and automated extraction of geometric and compositional features that are both experimentally robust and mechanistically informative. In particular, average covered area, droplet size distributions, and volume fraction serve as useful quantitative proxies for phase behavior. Such metrics are widely used in colloid science and formulation engineering to assess emulsion stability, encapsulation efficiency, and miscibility, and they allow us to build continuous, data-rich phase diagrams, rather than simple binary phase diagrams.

To demonstrate progress toward more thermodynamically grounded analyses, we have now added Supplementary Figure S26 (see response to Comment 1) that reports automated dense phase concentration measurements. This analysis leverages the same segmentation pipeline with slight modifications, which allowed us to extend our framework to extract more complex parameters without compromising throughput.

“To further extend the platform’s scope, we also performed preliminary measurements of dense-phase concentrations (Supplementary Figure S26), providing a basis for future composition-dependent analyses that extract partitioning coefficients and the associated thermodynamic driving forces.⁶²”

Finally, we see the current geometric and intensity-based descriptors not as endpoints, but as a foundation. They allow us to screen broadly, identify relevant regions of interest, and set the stage for deeper, targeted follow-up using advanced biophysical tools in future work.

“By enabling broad, quantitative screening, these metrics lay the groundwork for deeper, targeted analyses using advanced biophysical tools (i.e., FRAP, microrheology, or optical trapping) to probe material properties such as viscosity, dynamics, or mechanical behavior.”

Comment 3:

Two peptides of opposing charges were investigated. While a good model to demonstrate the basic utility of the chosen approach, their over-simplistic phase behaviour fails to address typical problems of assessing phase behaviour in more complex compounds such those inferred from high viscosity or aggregate formation (i.e. <http://dx.doi.org/10.1016/j.molcel.2015.08.018> , Fig 1B). Scikit may score non-roundish

appearance of highly viscous condensates by an ML algorithm, but how did the authors suggest to navigate aggregation or glass transitions? Can the algorithm differentiate them from condensates? How does it handle aggregates below the 500 px threshold?

Response:

We agree that distinguishing condensates from aggregates or high-viscosity phases is a well-recognized challenge in the field. In our current implementation, any connected pixel cluster above the 500-pixel threshold is labelled as “phase separated”, irrespective of morphology or internal dynamics. As such, the algorithm does not explicitly differentiate condensates from aggregates or glass-like states. This limitation has now been more clearly addressed in the manuscript, and we have added representative examples of rare salt-induced aggregates detected in our higher-salt 3D planes (see figure below, now added as Supplementary Figure S48). In the revised manuscript, we have added the following clarification:

“Interestingly, some phase-separated regions at higher salt concentrations were identified (Figure 6D, 270° rotation), which result from salt-induced aggregate phases (Supplementary Figure S48). As our current analysis detects any contiguous fluorescent signal above the pixel threshold, these aggregates were classified as ‘phase separated’, regardless of internal structure or material state.”

Supplementary Figure S48: Representative confocal micrographs of salt-induced aggregates formed by poly-L-(lysine)₁₀₀ and poly-L-(aspartic acid)₂₀₀ at high NaCl concentrations. From left to right: the raw confocal image, the corresponding binary mask obtained by thresholding, and an overlay highlighting detected aggregates (in red). All samples contain 0.1 mM lysine and 2050 mM NaCl, with increasing aspartic acid concentrations: 4.0, 5.7, and 7.8 mM for samples 1–3, respectively. These images illustrate the morphological appearance of the salt-induced aggregates discussed in Figure 6D of the main text.

While our use of simple polypeptides avoids aging effects (see Supplementary Figure S15 of our previous work) compared to the systems shown in Figures 3 of Lin et al. (Mol Cell, 2015), we agree that more complex systems may require additional classification layers. Future extensions of our platform could integrate orthogonal readouts, such as FRAP or microrheology, to capture dynamic material properties over time. Additionally, fluorescent dyes such as ThT and Amytracker can report on the presence of β -sheets. However, these additions

would reduce throughput and require substantial further development, which lies beyond the scope of this proof-of-concept study. We believe that establishing a robust and reproducible classification framework—even with a simplified system—is a critical first step toward tackling these more complex behaviors in future iterations of the platform. This future direction is now explicitly included in the Outlook section:

“Looking ahead, complementary tools such as fluorescence recovery after photobleaching (FRAP), microrheology, partitioning measurements, or the use of structure-sensitive dyes (e.g., ThT, Amytracker) could be incorporated to quantify condensate diffusivity, viscosity, molecular enrichment, or the presence of β -structured aggregates, further deepening functional insights.”

Comment 4:

Overall sample consumption appears unforgivingly high: 150 μ L per well and 40-250 samples for a phase diagram resulting in 6 to 37 mL of total sample. Protein purification can be elaborate. Thus, a reasonable sample consumption is crucial for the system to be adopted, yet along recognized in the field. Protein amounts consumed per phase diagram should be reported for all diagrams. Also, are the authors aware of manuscripts where such amounts of material were available to investigate macromolecular phase separation. Concurrently, how would a 10, 100, or 1000 fold reduction in sample consumption be achievable? Why did the authors not explore these?

Response:

Sample consumption in our current setup is well within reasonable limits for synthetic polymer systems and materials sciences. A typical high-resolution phase diagram comprising 72 samples consumes approximately 3 to 5 mg of material per component (e.g., poly-lysine and poly-aspartic acid), as now reported in the figure captions of all phase diagrams in the main text. Importantly, the extent of sample usage is inherently determined by the researcher’s choice of chemical space: in our study, we deliberately explored up to 8 mM monomer concentrations, resulting in low milligram-scale sample consumption. While this enabled a comprehensive phase diagram mapping of the system, it is not strictly necessary. Researchers interested in narrower concentration windows could restrict sampling accordingly and thus further reduce material demands.

We acknowledge, however, that sample consumption becomes more critical for proteins that are difficult to express or purify. Below, we suggest several opportunities to reduce sample usage through both experimental and computational strategies:

- Smaller batch size. One could sample fewer samples at a time to increase the efficiency of active learning.
- Early stopping. In practice, many phase diagrams can be stopped once sufficient coverage or accuracy is achieved, especially with active learning.
- Miniaturization. Switching from 96-well to 384-well formats could be explored to reduce per-condition volumes from 150 μ L to ≤ 30 μ L. This 5-fold reduction is particularly useful for high-dimensional diagrams or rare proteins.
- Sparse sampling. We systematically evaluated coarser sampling grids by selecting every third or fourth point. This yielded 9- to 15-fold reductions in the number of conditions in 2D (e.g., 6561 \rightarrow 729 or 441) and 27- to 53-fold reductions in 3D (e.g.,

531,441 \rightarrow 19,683 or 9,261). As shown in the figure below (now included as Supplementary Figure S49), the overall quality of these diagrams remains high, particularly in early active learning cycles. Precision at the phase boundary is slightly reduced but sufficient for most exploratory purposes.

Together, these examples show that both the sampling resolution and material usage per condition can be tuned, depending on the user's goals and constraints. We therefore view sample consumption as a flexible parameter, rather than a fixed limitation of the method. At the same time, the ability to extract rich kinetic information (Comment 4, Reviewer 5) and the potential to integrate additional biophysical tools (Comments 1–3, Reviewer 3) further increase the value of each sample. We now explicitly report the total mass used for each phase diagram in the respective figure captions and have added the figure below as Supplementary Figure S49.

Supplementary Figure S49: Effect of search space down-sampling on in silico active learning (AL) performance. **(A)** Balanced accuracy per AL cycle in 2D simulations using the default search space (6561 points, solid purple line), a 3 \times down-sampled grid (729 points, blue dash-dotted line), and a 4 \times down-sampled grid (441 points, black dash-dotted line). **(B)** Balanced accuracy per AL cycle in 3D simulations using the default grid (531,441 points, solid purple line), a 3 \times down-sampled grid (19,683 points, blue dash-dotted line), and a 4 \times down-sampled grid (9,261 points, black dash-dotted line). **(C)** Predicted 2D phase diagrams after 72 AL samples for the default (left), 3 \times down-sampled (middle), and 4 \times down-sampled (right) search spaces. Phase-separating conditions are shown in blue, non-separating conditions in red, and sampled points as black circles.

Comment 5:

BSA is used to passivate the surface of the well-plates. However, addition of BSA can also work as a crowding agent and thus influence phase separation behaviour. We would like to see a control experiment that this is not the case for the implemented protocol.

Response:

Surface passivation with BSA is a widely adopted method to suppress nonspecific adsorption in microplate-based assays and LLPS studies.^{13,14} In our protocol, the coating solution is removed after incubation and followed by three thorough washes with MQ water.

To confirm that any residual BSA does not influence phase behavior, we incubated BSA-coated wells with either water or experimental buffer and measured the supernatant using a Nanodrop spectrophotometer. As shown in panel A in the figure below (now added as Supplementary Figure S6), no absorbance at 280 nm was detected relative to a blank, indicating that any residual BSA in solution is below the detection threshold and present at negligible concentrations.

To further address this concern, we performed a conservative, worst-case estimate assuming complete desorption of a dense BSA monolayer from the well plate. Based on a surface area of 0.30 cm² and an estimated molecular footprint of ~44 nm², this corresponds to ~6.9 × 10¹¹ molecules or ~76 ng of protein per well. This is equivalent to 0.00076 mg/mL in 100 μL. To experimentally test this upper bound, we prepared a control sample with 0.001 mg/mL BSA and acquired a full phase diagram under identical conditions. No differences were observed (see panels B–C).

Together, these control experiments and calculations confirm that residual BSA, if present at all, does not measurably affect phase separation under our conditions.

Supplementary Figure S6: Assessment of BSA carryover effects on phase behavior. (A) Nanodrop absorbance at 280 nm for solutions incubated in BSA-coated wells for 15 minutes with either ultrapure water (left) or experimental buffer (right), compared to a blank and a 0.1 mg/mL BSA reference. No absorbance was detected above background, indicating negligible BSA desorption. (B–C) Phase diagrams of poly-L-(lysine)₁₀₀ and poly-L-(aspartic acid)₂₀₀ acquired without (B) or with (C) 0.001 mg/mL BSA supplemented in the bulk solution. This concentration reflects a conservative upper bound assuming complete BSA desorption from the plate. No differences in phase behavior were observed.

Comment 6:

The authors used a custom tip with a „touch“ function. To allow for accurate reproducibility all essential methodological details must be communicated clearly. We thus like would ask the authors to provide an explanation of the exact tip function in the method section.

Response:

The standard Opentrons “touch-tip” function involves contacting the well wall at up to four fixed positions (top, bottom, left, and right) to remove residual droplets after dispensing. While sufficient for simple transfers up to four liquids, this default behavior does not distinguish between reagents when multiple liquids are dispensed into the same well, which can lead to positional overlap and cross-contamination. To address this, we implemented a dynamic touch-tip routine that adapts both the number and location of contact points based on the number of liquids added per well. For example, in a 6-liquid system, six evenly spaced points (at 60° intervals) are assigned around the circular well perimeter, so each liquid contacts a distinct location. This feature ensures spatial separation, enabling accurate and reproducible multi-liquid dispensing without residual contamination.

We have now expanded the Methods section to clearly describe this custom routine and its role in improving dispensing precision and reproducibility:

“Custom Dispensing Technique. To improve accuracy, transfers used a minimum of 10 μL , leaving 5 μL of residual volume in the tip. Prior to dispensing, dynamic volume tracking adjusted the pipette height based on the anticipated liquid volume in the well. Dispensing occurred just above the liquid surface to ensure that any residual droplets hanging from the tip were reliably released into the bulk solution (Figure 1C, side view). After dispensing, a custom touch-tip function guided the pipette to contact specific points along the well wall at the same height to remove remaining droplets. Unlike the default four-point Opentrons routine, our implementation dynamically adjusted both the number and location of contact points based on the number of distinct liquids added to each well (Figure 1C, top view). As additional liquids were dispensed, new touch points were assigned at progressively higher vertical positions, forming a gentle upward spiral (Figure 1C, touch point trajectory). This approach enabled accurate multi-liquid dispensing with a single tip, minimized cross-contamination risk, and maintained spatial separation between touch locations regardless of the number of liquids used.”

We hope these added methodological details address the reviewers’ concern and improve the clarity and reproducibility of our protocol.

Reviewer #5:

Summary:

I have reviewed the manuscript titled “Automated navigation of condensate phase behavior with active machine learning,” which demonstrates how a data driven, iterative, automated method can efficiently explore material properties by targeting experiments to increasingly informative data points. The approach uses previous iterations to predict conditions that maximize information gain for mapping binary phase diagrams, achieving higher statistical power per measurement than conventional mapping approaches.

Response:

We thank the reviewer for their thoughtful and constructive evaluation, and we appreciate the time they took to provide this review. We are pleased that the core strengths of our data-driven, iterative approach were recognized, and we are grateful for their contribution to the assessment of our work.

Strengths

- 1) This work details a novel way of using automation and inference to improve the direction of experiments without requiring major changes to the assays and measurements commonly used to study coacervates. This approach has broad applicability across the coacervate research community and potentially to other domains where iterative experimentation is feasible.
- 2) The implementation utilizes accessible instrumentation to achieve significant advancements in experimental orchestration. The demonstration of dynamic, data-driven research—where collected data guides subsequent experimentation rather than following a predetermined protocol—represents a methodological paradigm that could be readily adopted by other research groups.
- 3) Specific to coacervates, this methodology can improve time efficiency, lower resource costs and improve the results for analysis of many liquid-liquid phase separating systems. Pieces of this methodology can be adapted with minimal effort, for example by replacing optical microscopy with 96 well plate DLS systems.
- 4) The active machine learning approach is an important demonstration, and the mathematical methods and model training are well conceived and well executed.

Response:

Many of the strengths noted by the reviewer indeed reflect core priorities that guided the development of this platform. We deliberately focused on making the system as accessible and adaptable as possible, with the hope that it could integrate into existing workflows without requiring major changes. It's very encouraging to see that these intentions come across clearly.

Weaknesses

- 1) While analytically powerful, the demonstrated applications remain confined to methodological validation. Despite showing capabilities for assaying various polymer lengths and salt concentrations, the approach was not extended to identify materials with novel properties or scientific significance, thus limiting the manuscript's impact.

We fully agree that extending this approach toward the discovery of materials with novel or application-specific properties is an exciting and important direction. In many ways, we see the current work as a foundational step toward developing a platform that not only maps phase diagrams, but actively guides the design of materials with tailored properties. While such applications lie beyond the scope of the present study, we are enthusiastic about this next phase. To reflect this direction, we now highlight the following in our outlook:

“Moreover, integrating condensate properties into active machine learning algorithms will allow us to incorporate desirable particle properties in the decision-making process, supporting the design of biomaterials for applications such as drug delivery and tissue engineering.”

- 2) The method's dependence on fluorescent peptide labeling restricts the range of coacervate-forming materials that can be investigated and potentially alters the intrinsic properties of the condensates being studied.

Following the reviewer's suggestion, we carried out additional control experiments (Comments 1 and 2; Reviewer 5) using label-free techniques (DLS and turbidity), which helped strengthen the orthogonal validation of our fluorescence-based measurements. We observed good alignment across methods, suggesting that covalent fluorescent labeling did not substantially affect the observed phase behavior. This aligns with our efforts to limit possible labeling effects, including the use of labeled polymers that were chemically identical to their unlabeled counterparts (Cy5-Lys20, Cy5-Lys100, and Cy5-Lys250), low labeling densities (<1 dye per polymer), and total dye concentrations below 250 nM.

Looking ahead, a key strength of the platform is its flexibility toward alternative detection strategies. Depending on the system, small-molecule dyes that selectively partition into coacervates (e.g., via hydrophobic interactions) could be used to visualize condensates without chemically modifying the coacervating components. In addition, the platform can be readily extended to incorporate high-throughput, label-free methods such as 96-well DLS, as also suggested by the reviewer. Together, these adaptations would expand compatibility to systems not amenable to covalent labeling and further broaden the range of accessible materials.

- 3) All experiments were conducted within established concentration ranges using well-characterized coacervate systems. The manuscript lacks verification of how effectively this approach would perform when determining phase boundaries for novel coacervate systems, particularly where the appropriate experimental resolution and range are not known a priori. Additional testing with less characterized materials would strengthen claims regarding the method's generalizability, especially for systems with concentration-dependent behaviors spanning wider ranges or exhibiting highly localized sensitivity within the parameter space.

Indeed, selecting an appropriate concentration range can be particularly challenging for novel or poorly characterized systems, where phase behavior is largely unknown. This platform is designed to remain effective even under such uncertainty. Users can define a broad initial search space based on solubility limits or estimated experimental feasibility, and the pipetting setup accommodates this flexibility without additional complexity. Importantly, the active learning algorithm is well-suited to this context. Within the user-defined bounds, it adaptively identifies the most informative regions of the parameter space, reducing the need for

exhaustive trial-and-error screening. This data-driven refinement enables the efficient construction of phase boundaries without requiring prior knowledge of the system's behavior.

This adaptive behavior is particularly advantageous in cases with narrow phase transitions or non-monotonic behavior, where key features may be easily missed using fixed sampling strategies. To demonstrate the platform's robustness under such conditions, we conducted an *in-silico* experiment in which negative-phase regions were artificially introduced into a previously learned phase diagram (see figure below, now added as Supplementary Figure S8). Despite the added complexity, the active learning algorithm rapidly adjusted its sampling strategy and successfully identified the newly introduced phase boundaries.

This illustrates the system's capacity to detect and resolve unexpected or localized variations in phase behavior without prior knowledge of their existence. In addition to the machine learning framework, the platform also benefits from high-resolution imaging capabilities. Specifically, the integrated confocal microscope offers improved spatial resolution compared to commonly used epifluorescence setups^{11,15}, making it more suitable for detecting small or early-stage condensates. This enhances detection robustness, particularly in systems where marginal phase separation might otherwise go unnoticed. Taken together, these features make the platform well-suited for exploratory studies and broadly applicable to less-characterized or more complex coacervate systems, including those with wider concentration ranges or sharply localized transitions, as noted by the reviewer.

We have now added Supplementary Figure S8 and added the following to our Main Text:

“As a further validation, we tested the pipeline on a synthetic phase diagram containing multiple isolated negative phases (Supplementary Figure S8). The model successfully identified these hidden regions, demonstrating its flexibility and robustness in navigating complex phase landscapes.”

Supplementary Figure S8: *In-silico* validation of the active machine learning framework for resolving complex phase landscapes. To evaluate the platform's ability to detect unexpected or highly localized phase boundaries, we constructed an artificial phase diagram with increased complexity. **(A)** The evolving probability landscape over 15 cycles demonstrates how the model progressively detects and outlines distinct phase regions separated from the main boundary. **(B)** The corresponding uncertainty maps illustrate a consistent reduction in prediction entropy across cycles, reflecting improved model certainty. Color scales for both panels are provided in Supplementary Figure S3. These results demonstrate that the active machine learning framework remains effective even when challenged with more complex and fragmented phase landscapes, successfully resolving subtle and spatially separated features.

Points Requiring Revision

Comment 1:

The binary coacervate forming criteria used is somewhat arbitrary, making the interpretation of the phase diagrams problematic. This is highlighted in Figure 2B, pane 2 where large numbers of particles are seen but are likely excluded by the seemingly arbitrary 500-pixel criteria for including a particle. Panes 3 and 4 also show significant numbers of particles and it is unclear why the negative and positive example images are presented with different background intensities. Confirmation of these results with volumetric measurements like turbidity or DLS should be performed to ensure alignment of this criteria with the more common standards in the literature. As an example, if Pane 2 in Figure 2B has large concentrations of small, phase separated coacervates they may be missed by size or reduced settling.

Response:

Below, we address the reviewer's concerns regarding image contrast, particle size thresholds, and validation using volumetric techniques.

First, to clarify the contrast differences in Main Text Figure 2B, we have prepared a side-by-side comparison below (now added as Supplementary Figure S2) showing the same images with either individually optimized contrast (as used in the manuscript) or a single global contrast applied across all panels. As the figure illustrates, applying one contrast level renders the negative samples (panes 1–4) almost entirely black, making it difficult to visually assess the presence or absence of droplets. To avoid this, we applied contrast optimization per image to ensure that low background signals remain visible, allowing a more accurate and transparent interpretation. All images, including the positive samples (panes 5–8), were adjusted using this approach. Notably, this also helps to minimize subtle differences in droplet brightness, such as in pane 5, where reduced dye accumulation leads to lower signal intensity. Our aim in this figure was to show whether or not phase-separated particles are present, and we believe that optimizing the contrast in each image is the fairest and clearest way to support that objective.

Supplementary Figure S2: Effect of contrast scaling on the interpretability of confocal images.

Left: Images with individually optimized contrast to enhance visibility and distinguish negative (1–4) from positive (5–8) samples. This approach was used in the main manuscript to maximize contrast in each image. *Right:* Same images with uniform global contrast. While preserving raw intensity values, this renders panes 1–4 nearly black and visually uninformative. Additionally, pane 5 appears dimmer due to reduced dye accumulation in these droplets.

Second, regarding the 500-pixel size cutoff used in our automated image segmentation; in any fully automated classification workflow, it is essential to define a lower size threshold to differentiate true condensate droplets from background variation, small dye aggregates, or imaging noise. We set this threshold at 500 pixels to ensure consistent and conservative detection, reducing the likelihood of false positives. This is particularly important in the context of active learning where misclassifying noise as droplets would cause the model to repeatedly

sample uninformative regions in subsequent iterations, undermining both learning efficiency and interpretability. While this approach may exclude very small puncta, such as those (barely distinguishable) in pane 2, we consider this a necessary and fair trade-off for maintaining robustness and scalability across the full experimental space.

To independently assess the accuracy of our classification approach, we performed complementary turbidity and dynamic light scattering (DLS) measurements on a subset of formulations near the predicted phase boundary (see figure below, now added as Supplementary Figure S9). These experiments confirmed the presence of nanometer scale particles at low poly-*L*-(aspartic acid) concentrations. As anticipated, these fall below the resolution threshold of our automated confocal microscopy pipeline and are therefore not detected. At higher concentrations, we observed a clear transition to micrometer sized droplets, consistent with our original phase assignments. While turbidity and DLS offer useful complementary validation, both have notable limitations: turbidity lacks spatial resolution and yields limited quantitative insight, whereas DLS is optimized for small particles but suffers from sedimentation artifacts and is fundamentally incompatible with larger condensates above 10 micrometers. In contrast, our confocal microscopy platform enables spatially resolved, high throughput classification and property quantification across hundreds of conditions, providing a more robust and informative tool for mapping condensate phase behavior.

Supplementary Figure S9: Characterization of poly-*L*-(lysine)₁₀₀ / poly-*L*-(aspartic acid)₂₀₀ formulations near the predicted phase boundary via turbidity and dynamic light scattering (DLS). (A) Zoom-in of the analyzed region within the phase diagram. The background gradient corresponds to the phase probability of the “ground-truth” phase diagram used in the main text (Figure 4). Experimental formulations were prepared along the dotted line at a fixed 2.0 mM of poly-*L*-(lysine)₁₀₀ with varying of poly-*L*-(aspartic acid)₂₀₀ concentrations (0.01–1.0 mM) in 50 mM HEPES (pH 7.4) with 150 mM NaCl. The approximate onset of phase separation along this trajectory was identified by the automated analysis workflow at around 2.0 mM lysine and 0.5 mM aspartic acid (black dot). (B) Turbidity ($\lambda = 600$ nm) as a function of poly-*L*-(aspartic acid)₂₀₀ concentration. Values remain low at concentrations (<0.1 mM) and increase sharply above 0.5 mM, indicating the onset of phase separation. (C) DLS-derived Z-average hydrodynamic diameters for the same formulations as in (B). Nanometer-scale particles (~170–210 nm) are detected between 0.05 and 0.1 mM poly-*L*-(aspartic acid)₂₀₀, while micrometer-scale droplets are observed above 0.5 mM. All measurements were performed 15 minutes after sample preparation, consistent with the timing used throughout the automated experimental pipeline. Data represent mean \pm SD ($n = 3$).

In summary, we have clarified the use of individually optimized contrast settings to enable fair visual comparison across samples and explained the rationale for applying a conservative 500-pixel size threshold to ensure reliable, noise-resistant classification in a fully automated workflow. While this inevitably excludes very small puncta near the detection limit, it allows consistent analysis across hundreds of conditions without compromising interpretability. Supporting this approach, independent turbidity and DLS measurements show good

agreement with our classifications, particularly for samples containing micrometer-sized droplets. More broadly, across all techniques, the question remains where to draw the line between background assemblies and true (liquid-liquid) phase-separated condensates. Our method takes a practical and conservative stance on this question, trading sensitivity at the nanoscale for robustness and scalability. Besides Supplementary Figure S2 and S9, we also added the following to our manuscript:

“Additionally, turbidity and DLS measurements support the observed phase boundary (Supplementary Figure S9), although DLS also detected the formation of nanometer-scale assemblies below the resolution limit of confocal microscopy.”

Comment 2:

The manuscript gives no details about the timing from formation of coacervates to the measurement of particles, even though both settling time and coalescence of particles represent dynamic events. Without that information, it is difficult to judge the reliability of the quantification criteria used. Additional experiments to confirm that these measurements are consistent with other coacervate quantification methods using the same timing should be done.

Response:

All coacervate samples in this work were imaged approximately 15 minutes after initial mixing. This incubation time reflects typical manual workflows, where similar delays often occur between sample preparation and image acquisition. It also aligns with prior reports on coacervate formation kinetics, where comparable timeframes are used to assess the presence and morphology of condensates.^{16–18} Based on our experience, 15 minutes is generally sufficient to determine whether coacervates have formed.

However, we agree with the reviewer that processes such as sedimentation and coalescence likely continue beyond this point, particularly for nanoscale coacervates. To evaluate whether this incubation time captures a representative stage of coacervate development, we performed time-resolved turbidity and dynamic light scattering (DLS) measurements across multiple formulations. These data, now included as Supplementary Figure S25, show that for formulations containing 2.0 mM poly-*L*-(lysine)₁₀₀ and varying poly-*L*-(aspartic acid)₂₀₀, turbidity increases rapidly and stabilizes after approximately 10 minutes. Similarly, time correlation functions from DLS measurements (panels B-D) indicate that the coacervates stabilize around the 15-minute mark. Micron-sized particles likely form due to the coalescence of nanometric coacervates, as indicated by the correlation coefficient increase beyond 10³ μs. Such coacervate evolution is especially remarkable for the samples with the highest poly-*L*-(aspartic acid)₂₀₀ concentrations (panels C-D).

Supplementary Figure S25: Time-resolved measurements validating the 15-minute incubation time used in the automated pipeline. (A) Turbidity kinetics of poly-*L*-(lysine)₁₀₀ / poly-*L*-(aspartic acid)₂₀₀ formulations near the phase boundary. Poly-*L*-(lysine)₁₀₀ was fixed at 2.00 mM, while poly-*L*-(aspartic acid)₂₀₀ was varied from 0.1 to 1.00 mM. (B–D) Time correlation functions obtained by dynamic light scattering (DLS) for the same formulations, measured between 0 and 25 minutes post-mixing (timepoints indicated in the legend). The decay of the autocorrelation function indicates the diffusion behavior of the coacervates, which is linked to particle size via the Stokes-Einstein equation. These correlograms, particularly in panels C and D, reveal a progressive flattening that reflects changes in particle size distribution and droplet growth over time. This flattening plateaued around 15 minutes for each sample, supporting the use of the 15 minute-mark as a consistent and practical snapshot for coacervate quantification. Polypeptide concentrations are reported in monomer concentrations.

Taken together, these results indicate that while the system continues to slightly evolve beyond 15 minutes, as is typical for coacervate systems, this time point provides a consistent and operationally practical snapshot for comparative analysis across conditions. The 15-minute incubation time is now explicitly stated in the Methods section:

“Samples were incubated for 15 minutes before analysis, unless indicated otherwise.”

Importantly, in our automated setup, this parameter can be readily adjusted to suit different sample types or user-defined experimental requirements (see Comment 4).

Comment 3:

The manuscript states “Some phase-separated regions at higher salt concentrations were identified (Figure 6D, 270° rotation), resulting from salt-induced aggregate phases.” Images from this positively scored coacervate region are not provided or discussed for image analysis, however examples from other reports show that protein aggregates are highly distinct from

condensates in optical images. Understanding how the protein aggregate images result in being scored as positive for condensates is not addressed.

Source: Peptide-based coacervates for enhanced function through structural organization and substrate specificity. Nat Commun. 2024 Oct 30;15(1):9368. doi: 10.1038/s41467-024-53699-z. Erratum in: Nat Commun. 2024 Nov 22;15(1):10125. doi: 10.1038/s41467-024-54617-z. PMID: 39477955; PMCID: PMC11525812.

Response:

These high-salt samples were scored as “positive” in our classification because our current image analysis pipeline defines phase separation based on the presence of localized, dye-enriched features above a defined size and intensity threshold, regardless of their underlying morphology. As such, both condensates and salt-induced aggregates can trigger a positive classification if they meet these criteria. This limitation has now been more clearly addressed in the manuscript, and we have added representative examples of rare salt-induced aggregates detected in our higher-salt 3D planes (see figure below, now added as Supplementary Figure S48). In the revised manuscript, we have added the following clarification:

“Interestingly, some phase-separated regions at higher salt concentrations were identified (Figure 6D, 270° rotation), which result from salt-induced aggregate phases (Supplementary Figure S48). As our current analysis detects any contiguous fluorescent signal above the pixel threshold, these aggregates were classified as ‘phase separated’, regardless of internal structure or material state.”

Supplementary Figure S48: Representative confocal micrographs of salt-induced aggregates formed by poly-*L*-(lysine)₁₀₀ and poly-*L*-(aspartic acid)₂₀₀ at high NaCl concentrations. From left to right: the raw confocal image, the corresponding binary mask obtained by thresholding, and an overlay highlighting detected aggregates (in red). All samples contain 0.1 mM lysine and 2050 mM NaCl, with increasing aspartic acid concentrations: 4.0, 5.7, and 7.8 mM for samples 1–3, respectively. These images illustrate the morphological appearance of the salt-induced aggregates discussed in Figure 6D of the main text.

While distinguishing between condensates and aggregates may be straightforward by visual inspection of individual images, implementing a fully automated, generalizable method for this distinction, robust across a wide range of morphologies and imaging conditions, is a nontrivial task and remains an area of active development. Such functionality was beyond the scope of the current work, which focused on robust and scalable detection of phase-separated domains rather than their mechanistic origin. We hope this clarification, along with the newly added figure, addresses the reviewer's concern and provides helpful context for interpreting the classifications in these conditions.

Comment 4:

The measurements of number of condensates, average particle area and total volume fraction should be demonstrated to be self-consistent over time. It is impossible to tell from the data if the snapshot captured represents a stable state or a moment in a continuously evolving system due to particle settling and coalescence. As a result, this cannot be interpreted rigorously.

Response:

We fully agree with the reviewer that condensate systems are dynamic, and that properties such as particle number, area, and volume fraction are influenced by time-dependent processes including sedimentation and coalescence.

To investigate this in more detail, we used our automated platform to perform time-resolved experiments with precisely controlled incubation times. As shown in the figure below (now added as Supplementary Figure S22), we mapped phase diagrams at incubation times of 5, 15, 30, and 60 minutes, with no user intervention during this period. Each diagram was

generated as a separate experiment, with tailored formulation sets selected to capture the evolving phase boundaries. At early time points (e.g., 5 minutes), many coacervates remain too small or insufficiently settled to be detected in the imaging plane, resulting in an underrepresentation of phase separation. However, from 15 minutes onward, the mapped phase boundaries remain consistent across timepoints, in line with turbidity and DLS data (see Reviewer 5, Comment 2).

Supplementary Figure S22: Time-resolved phase diagram mapping of poly-L-(lysine)₁₀₀ / poly-L-(aspartic acid)₂₀₀ coacervates. Phase diagrams were constructed by imaging after incubation times of 5, 15, 30, and 60 minutes. Each condition was treated as a separate experiment. The operational phase boundaries are shown as dashed lines for each time point. Substantial shifts in the phase boundary are observed between 5 and 15 minutes, indicating ongoing droplet settling. Beyond 15 minutes, the phase boundary remains stable, suggesting that the system has reached a near-equilibrium state under the experimental conditions. Polypeptide concentrations are reported in monomer concentrations.

We also imaged samples over an extended time course to quantify how condensate properties evolve after initial mixing (see figure below, now added as Supplementary Figure S23). From the confocal micrographs in panel A, we observe a similar trend where progressively more and larger condensates appear over time, particularly within the first 15 minutes. Panels B–D further illustrate that our measurements reflect snapshots of a continuously evolving system, where condensate properties change gradually due to ongoing fusion and sedimentation. As expected, the absolute values of particle count, area, and intensity depend on the selected incubation time. For instance, we observe an initial increase in particle count, followed by a plateau or slight decline, consistent with droplet fusion over time. Mean intensity initially increases due to sedimentation of condensates moving into the imaging plane, and then gradually decreases, primarily due to photobleaching caused by repeated and prolonged imaging. These trends align with the physical expectations for coacervate systems, and we find it encouraging that our platform can capture such behavior consistently and quantitatively across timepoints.

Supplementary Figure S23: Time-resolved mapping of coacervate properties in poly-L-(lysine)₁₀₀ and poly-L-(aspartic acid)₂₀₀ formulations. (A) Representative confocal micrographs of three selected formulations containing 2.0 mM poly-L-(lysine)₁₀₀ combined with 0.5, 1.0, or 2.0 mM poly-L-(aspartic acid)₂₀₀. The shown micrographs were taken at approximately 1, 15, 30, and 60 minutes after mixing. (B–D) Quantitative analysis of condensate properties for the same formulations over time: total particle count (B), mean particle area (C), and mean fluorescence intensity (D). These measurements reveal characteristic LLPS dynamics, including initial coalescence, settling into the imaging plane, and time-dependent photobleaching of fluorescent signal due to repeated imaging. Polypeptide concentrations are reported in monomer concentrations.

In summary, we chose an incubation time of 15 minutes for the present study, as it provides a practical balance between detection performance and measurement throughput. This parameter is fully configurable and can easily be extended, if desired, towards much longer timescales, depending on the user's objectives. We now explicitly state in the main text that all measurements represent defined-timepoint snapshots, and we have added Supplementary Figures S22 and S23 to transparently demonstrate how incubation time influences both phase boundary assignment and condensate property measurements.

“Notably, our property measurements represent standardized snapshot observations taken after a 15-minute incubation period of a dynamic, continuously evolving system. While the incubation time can be adjusted to suit the user's objective or extended to enable kinetic measurements (Supplementary Figure S22–24), we selected 15 minutes as a practical and reproducible readout, based on DLS and turbidity data showing that key features for identifying phase separation begin to stabilize around this time (Supplementary Figure S25).”

Comment 5:

Similarly, example segmentation of condensate confocal images is not provided. The varied confocal slice thickness from the glass to 50 μm above the glass and the overlapping of particles in images poses significant challenges for accurate quantification of condensate properties. Without being able to review that data, these measurements cannot be validated for accuracy. Based on these limitations, the condensate properties measured for particle count, average area and volume fraction cannot be verified from this data as presented.

Response:

Example segmentations are shown in Main Text Figure 1E; however, we recognize that this was not clearly stated in the original figure caption. We have now revised the caption to explicitly clarify that the segmentation shown is based on real confocal microscopy data.

“(E) Example segmentation of representative confocal microscopy data using automated binary Yen-thresholding for particle detection in each Z-plane.”

To further address this concern, we have also compiled a set of representative segmentation examples across different imaging depths and conditions, which are now included as Supplementary Figure S1. We hope these additions provide sufficient transparency regarding the segmentation quality and help validate the accuracy of the measured condensate properties.

Supplementary Figure S1: Representative confocal micrographs, segmentation, and particle detection of varying condensate forming conditions of poly-L-(lysine)₁₀₀ and poly-L-(aspartic acid)₂₀₀. From left to right: the raw confocal image, the corresponding binary mask obtained by Yen-thresholding, and an overlay highlighting detected coacervates above the 500 pixel threshold (in red). All samples contain 150 mM NaCl, with increasing concentrations of aspartic acid (2.5, 4.1, 8.1 mM), lysine (3.7, 6.0, 7.5 mM), and imaging depths of 1, 3, and 7 μm for samples 1–3, respectively. These images depict the segmentation and particle detection methods employed in the automated image analysis throughout all experiments.

Comment 6:

Based on the limitations outlined above, the reported condensate properties (particle count, average area, and volume fraction) require independent verification. Additional confirmatory measurements using complementary techniques such as DLS or other scattering methods sensitive to all phase interfaces are necessary to validate these measurements. If these properties are not integral to the active learning process and have not been rigorously validated, this data should either be substantiated with additional evidence or removed from the manuscript.

Response:

To provide additional validation of the condensate properties reported in our manuscript, we conducted independent measurements using nanoparticle tracking analysis (NTA). NTA enables visualization and quantification of nanoparticles in solution by tracking their Brownian motion using laser light scattering microscopy. This technique is well-suited for detecting particles in the 10 to 1000 nanometer range and offers a direct estimate of particle concentration in dilute nanometric systems.

As shown in the figure below (now added as Supplementary Figure S21), the mean particle sizes measured by NTA align with the nanosized clusters detected with DLS. Importantly, NTA allowed us to count the number of coacervates tracked in the observation volume. These measurements were used to estimate coacervate volume fractions across the formulation space. While NTA is particularly effective at detecting small, dilute coacervates in the nanometer regime, it is less suited for larger coacervates, leading to substantial propagated errors in volume fraction estimates at higher concentrations of poly-*L*-(aspartic acid)₂₀₀. In contrast, our confocal imaging pipeline is optimized for quantifying micrometer-sized droplets and more concentrated systems. At intermediate concentrations, such as the 1.0 mM poly-*L*-(aspartic acid)₂₀₀ condition where both techniques are operational, we observed volume fraction estimates that were of the same order of magnitude (see main text, Figure 4; e.g., 0.004% for a formulation with 1.3 mM poly-*L*-(aspartic acid)₂₀₀ and 2.5 mM poly-*L*-(lysine)₁₀₀), consistent with the range obtained by NTA.

Supplementary Figure S21: Nanoparticle Tracking Analysis (NTA) of coacervates formed from poly-L-(lysine)₁₀₀ / poly-L-(aspartic acid)₂₀₀. (A–C) Particle size distributions showing concentration per size bin for samples containing: (A) 0.1 mM, (B) 0.5 mM, and (C) 1.0 mM poly-L-(aspartic acid)₂₀₀. In all cases, poly-L-(lysine)₁₀₀ was fixed at 2.0 mM. (D) Summary of key coacervate properties as determined by NTA, including mean particle size, particle concentration, estimated single-particle volume, and total coacervate volume fraction per condition. Polypeptide concentrations are reported in monomer concentrations.

In summary, we have provided complementary validation of volume fraction estimates using NTA. While NTA is limited in its ability to quantify larger coacervates, our imaging-based platform is specifically designed to capture such features. These property measurements represent a core strength of the system and offer a solid foundation for future extensions. In particular, they highlight the potential of the platform to support more advanced and multi-objective optimization strategies, which we consider an important next step toward the discovery of new functional materials.

Comment 7:

This research makes a significant contribution by introducing a more efficient, automated approach to mapping coacervate phase boundaries and demonstrating a data-driven experimental methodology that strategically targets formulation parameters to maximize the information value of each collected data point. With the suggested revisions, I believe this work would constitute a valuable contribution to Nature Communications.

Response:

We sincerely appreciate the reviewer's thoughtful and constructive feedback, as well as the time and effort dedicated to evaluating our manuscript. Their comments helped us to clarify key aspects and strengthen the validation of our results through orthogonal measurements. We are also grateful for the reviewer's positive assessment and recognition of the manuscript's contribution to automated coacervate phase mapping and data-driven experimental design.

Reference list:

1. Scalia, G., Grambow, C. A., Pernici, B., Li, Y. P. & Green, W. H. Evaluating Scalable Uncertainty Estimation Methods for Deep Learning-Based Molecular Property Prediction. *J Chem Inf Model* **60**, 2697–2717 (2020).
2. Ortiz-Perez, A., van Tilborg, D., van der Meel, R., Grisoni, F. & Albertazzi, L. Machine learning-guided high throughput nanoparticle design. *Digital Discovery* **3**, 1280–1291 (2024).
3. Ribeiro, S. S., Samanta, N., Ebbinghaus, S. & Marcos, J. C. The synergic effect of water and biomolecules in intracellular phase separation. *Nature Reviews Chemistry* **2019** 3:9 **3**, 552–561 (2019).
4. Dignon, G. L., Best, R. B. & Mittal, J. Biomolecular phase separation: From molecular driving forces to macroscopic properties. *Annu Rev Phys Chem* **71**, 53–75 (2020).
5. Milin, A. N. & Deniz, A. A. Reentrant Phase Transitions and Non-Equilibrium Dynamics in Membraneless Organelles. *Biochemistry* **57**, 2470–2477 (2018).
6. Alberti, S., Gladfelter, A. & Mittag, T. Considerations and challenges in studying liquid-liquid phase separation and biomolecular condensates. *Cell* **176**, 419 (2019).
7. Von Bergen, M., Barghorn, S., Biernat, J., Mandelkow, E. M. & Mandelkow, E. Tau aggregation is driven by a transition from random coil to beta sheet structure. *Biochim Biophys Acta Mol Basis Dis* **1739**, 158–166 (2005).
8. Emmanouilidis, L. *et al.* A solid beta-sheet structure is formed at the surface of FUS droplets during aging. *Nat Chem Biol* **20**, 1044–1052 (2024).
9. Patel, A. *et al.* A Liquid-to-Solid Phase Transition of the ALS Protein FUS Accelerated by Disease Mutation. *Cell* **162**, 1066–1077 (2015).
10. Chatterjee, S. *et al.* Reversible Kinetic Trapping of FUS Biomolecular Condensates. *Advanced Science* **9**, (2022).
11. Arter, W. E. *et al.* Biomolecular condensate phase diagrams with a combinatorial microdroplet platform. *Nature Communications* **2022** 13:1 **13**, 1–10 (2022).
12. Posey, A. E. *et al.* Biomolecular Condensates are Characterized by Interphase Electric Potentials. *J Am Chem Soc* **146**, 28268–28281 (2024).
13. Hong, K., Song, D. & Jung, Y. Behavior control of membrane-less protein liquid condensates with metal ion-induced phase separation. *Nat Commun* **11**, 1–12 (2020).
14. Yao, R.-W. & Rosen, M. K. Advanced Surface Passivation for High-Sensitivity Studies of Biomolecular Condensates. *bioRxiv* 2024.02.12.580000 (2024) doi:10.1101/2024.02.12.580000.
15. Kopp, M. R. G. *et al.* Microfluidic Shrinking Droplet Concentrator for Analyte Detection and Phase Separation of Protein Solutions. *Anal Chem* **92**, 5803–5812 (2020).
16. Tian, L. *et al.* Spontaneous assembly of chemically encoded two-dimensional coacervate droplet arrays by acoustic wave patterning. *Nat Commun* **7**, 1–10 (2016).
17. Wu, X., Sun, Y., Yu, J. & Miserez, A. Tuning the viscoelastic properties of peptide coacervates by single amino acid mutations and salt kosmotropicity. *Communications Chemistry* **2024** 7:1 **7**, 1–12 (2024).
18. Karoui, H., Seck, M. J. & Martin, N. Self-programmed enzyme phase separation and multiphase coacervate droplet organization. *Chem Sci* **12**, 2794–2802 (2021).

Point-by-point response to the final reviewers' comments

Reviewer 5:
One lingering subject comes up from reviewing the authors' responses, though it does not require changes. If the system has the advantage of detecting larger coacervate droplets, but misses smaller, but significant quantities of smaller droplets, what is the significance or advantage of detecting larger droplets beyond extending the range of measurements above an NTA? Are larger droplets more physiologically relevant, etc. This brings in the definition of what a LLPS phase diagram is, and if it is determined from first principles then size limited measurements are not likely to yield results that match theoretical predictions. For the purpose of this publication, it is unnecessary to resolve this question, and inclusion of the orthogonal data allows an informed reader to interpret the experiments as presented. The addition of simulations to validate that the methodology for more complex phase boundaries is very encouraging and a great addition that demonstrates the potential beyond the repetitive (but meaningful) curves the manuscript focuses on. I look forward to the extension of this approach to other materials and believe it could be a phenomenal tool for first advancing first principle analysis, when phase boundaries are predicted and the automated systems can be used to validate or refute the predicted results or train a reasoning model. Tuning this system to investigate edge cases or identify the limits of theories could be as powerful as the ability to classify a large number of materials to establish a database.

>>Response:

Regarding droplet size: The reviewer raises an interesting conceptual point about the relative significance of detecting larger versus smaller condensates. Larger droplets are often associated with more mature or long-lived states of phase separation, and thus may be particularly relevant when linking phase behavior to downstream material properties. At the same time, smaller droplets can carry important biological meaning, for example in nucleation events or transient assemblies. Our current system relies on confocal microscopy and is therefore diffraction-limited; as such, smaller droplets may be underrepresented. We see our approach as a practical and scalable means of identifying conditions where condensate formation is unambiguous at the mesoscale. Rather than a substitute for techniques such as NTA or DLS, it provides a complementary readout that emphasizes throughput and robustness. Looking ahead, we are indeed also excited about the possibility of integrating orthogonal systems specialized in detecting small condensates into our pipeline.

Reviewer 5:

The discussion around comment 2 was probably the weakest response by the authors. From the reviewers closing comment:

“To our understanding important aspects of condensates are material properties like viscosity, diffusivity of proteins, partitioning coefficient or Gibbs free energy (see <https://doi.org/10.1038/s41587-019-0341-6> and <https://doi.org/10.1038/s41586-020-2256-2>). What specific value could the authors extract from these analysed covered area & volume fraction? “

The authors explain why they can't use three established techniques, then mention the partition coefficient provides thermodynamic parameters, but they don't demonstrate that or provide confirmation that their partition coefficients match any of the three established methods. Since this is unproven in a rigorous manner, they might want to lighten the claim. It seems like they are explaining why their method isn't authoritative but serves as a strong practical surrogate for better defined methods. That could be followed up on without high throughput.

Generally, their response to this comment is one of clarifying what they are claiming, but not necessarily just saying, no it isn't proven effective for that at the present time.

>>Response:

Regarding partition coefficients and thermodynamic parameters: We agree with the reviewer that our initial phrasing could be interpreted as overstating the extent to which our current

measurements provide thermodynamic information. To address this, we have revised the relevant sentence in the manuscript to lighten the claim:

“To further extend the platform’s scope, we also performed preliminary measurements of dense-phase concentrations (Supplementary Figure 26), providing a basis for future composition-dependent analyses that could approximate partitioning coefficients and the associated thermodynamic driving forces.”

This change makes clear that our measurements are intended as practical, high-throughput surrogates that can guide comparative mapping, rather than rigorous determinations of thermodynamic parameters.

Finally, we greatly appreciate the reviewer’s generous and forward-looking perspective on the potential of our approach. Their vision of how automated systems can ultimately be used to test theoretical boundaries and build comprehensive datasets resonates strongly with our own long-term goals.

Reviewer Comments to Author

I have reviewed the manuscript titled “Automated navigation of condensate phase behavior with active machine learning,” which demonstrates how a data driven, iterative, automated method can efficiently explore material properties by targeting experiments to increasingly informative data points. The approach uses previous iterations to predict conditions that maximize information gain for mapping binary phase diagrams, achieving higher statistical power per measurement than conventional mapping approaches.

Strengths

This work details a novel way of using automation and inference to improve the direction of experiments without requiring major changes to the assays and measurements commonly used to study coacervates. This approach has broad applicability across the coacervate research community and potentially to other domains where iterative experimentation is feasible.

The implementation utilizes accessible instrumentation to achieve significant advancements in experimental orchestration. The demonstration of dynamic, data-driven research—where collected data guides subsequent experimentation rather than following a predetermined protocol—represents a methodological paradigm that could be readily adopted by other research groups.

Specific to coacervates, this methodology can improve time efficiency, lower resource costs and improve the results for analysis of many liquid-liquid phase separating systems. Pieces of this methodology can be adapted with minimal effort, for example by replacing optical microscopy with 96 well plate DLS systems.

The active machine learning approach is an important demonstration, and the mathematical methods and model training are well conceived and well executed.

Weaknesses

While analytically powerful, the demonstrated applications remain confined to methodological validation. Despite showing capabilities for assaying various polymer lengths and salt concentrations, the approach was not extended to identify materials with novel properties or scientific significance, thus limiting the manuscript's impact. The method's dependence on fluorescent peptide labeling restricts the range of coacervate-forming materials that can be investigated and potentially alters the intrinsic properties of the condensates being studied.

All experiments were conducted within established concentration ranges using well-characterized coacervate systems. The manuscript lacks verification of how effectively this approach would perform when determining phase boundaries for novel coacervate

systems, particularly where the appropriate experimental resolution and range are not known a priori. Additional testing with less characterized materials would strengthen claims regarding the method's generalizability, especially for systems with concentration-dependent behaviors spanning wider ranges or exhibiting highly localized sensitivity within the parameter space.

Points Requiring Revision

1. The binary coacervate forming criteria used is somewhat arbitrary, making the interpretation of the phase diagrams problematic. This is highlighted in Figure 2B, pane 2 where large numbers of particles are seen but are likely excluded by the seemingly arbitrary 500-pixel criteria for including a particle. Panes 3 and 4 also show significant numbers of particles and it is unclear why the negative and positive example images are presented with different background intensities. Confirmation of these results with volumetric measurements like turbidity or DLS should be performed to ensure alignment of this criteria with the more common standards in the literature. As an example, if Pane 2 in Figure 2B has large concentrations of small, phase separated coacervates they may be missed by size or reduced settling.
2. The manuscript gives no details about the timing from formation of coacervates to the measurement of particles, even though both settling time and coalescence of particles represent dynamic events. Without that information, it is difficult to judge the reliability of the quantification criteria used. Additional experiments to confirm that these measurements are consistent with other coacervate quantification methods using the same timing should be done.
3. The manuscript states “Some phase-separated regions at higher salt concentrations were identified (Figure 6D, 270° rotation), resulting from salt-induced aggregate phases.” Images from this positively scored coacervate region are not provided or discussed for image analysis, however examples from other reports show that protein aggregates are highly distinct from condensates in optical images:

Source: Peptide-based coacervates for enhanced function through structural organization and substrate specificity. Nat Commun. 2024 Oct 30;15(1):9368. doi: 10.1038/s41467-024-53699-z. Erratum in: Nat Commun. 2024 Nov 22;15(1):10125. doi: 10.1038/s41467-024-54617-z. PMID: 39477955; PMCID: PMC11525812.

Understanding how the protein aggregate images result in being scored as positive for condensates is not addressed.

4. The measurements of number of condensates, average particle area and total volume fraction should be demonstrated to be self-consistent over time. It is impossible to tell from the data if the snapshot captured represents a stable state or a moment in a continuously evolving system due to particle settling and coalescence. As a result, this cannot be interpreted rigorously.

Similarly, example segmentation of condensate confocal images is not provided. The varied confocal slice thickness from the glass to 50 μm above the glass and the overlapping of particles in images poses significant challenges for accurate quantification of condensate properties. Without being able to review that data, these measurements cannot be validated for accuracy. Based on these limitations, the condensate properties measured for particle count, average area and volume fraction cannot be verified from this data as presented.

5. Based on the limitations outlined above, the reported condensate properties (particle count, average area, and volume fraction) require independent verification. Additional confirmatory measurements using complementary techniques such as DLS or other scattering methods sensitive to all phase interfaces are necessary to validate these measurements. If these properties are not integral to the active learning process and have not been rigorously validated, this data should either be substantiated with additional evidence or removed from the manuscript.

This research makes a significant contribution by introducing a more efficient, automated approach to mapping coacervate phase boundaries and demonstrating a data-driven experimental methodology that strategically targets formulation parameters to maximize the information value of each collected data point. With the suggested revisions, I believe this work would constitute a valuable contribution to Nature Communications.